# Intriguing Properties of Generative Classifiers

**Priyank Jaini**\*
Google DeepMind

**Kevin Clark**
Google DeepMind

**Robert Geirhos**\*
Google DeepMind

## Abstract

What is the best paradigm to recognize objects—discriminative inference (fast but potentially prone to shortcut learning) or using a generative model (slow but potentially more robust)? We build on recent advances in generative modeling that turn text-to-image models into classifiers. This allows us to study their behavior and to compare them against discriminative models and human psychophysical data. We report four intriguing emergent properties of generative classifiers: they show a record-breaking human-like shape bias (99% for Imagen), near human-level out-of-distribution accuracy, state-of-the-art alignment with human classification errors, and they understand certain perceptual illusions. Our results indicate that while the current dominant paradigm for modeling human object recognition is discriminative inference, zero-shot generative models approximate human object recognition data surprisingly well.

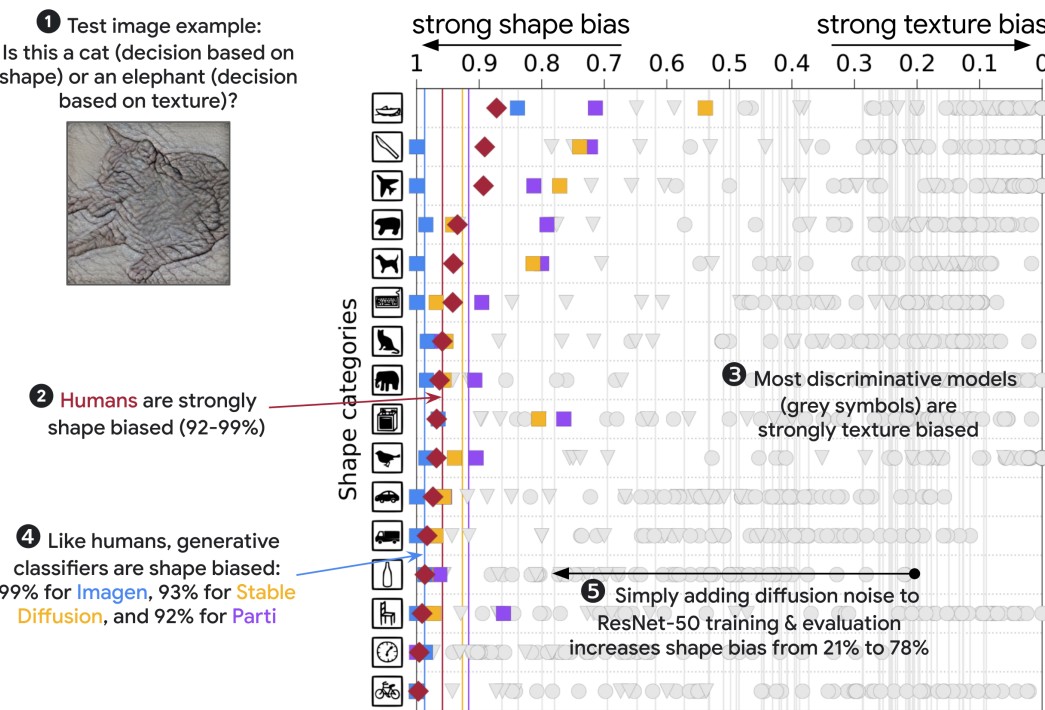

Figure 1: Zero-shot generative classifiers achieve a **human-level shape bias**: 99% for Imagen, 93% for Stable Diffusion, 92% for Parti and 92–99% for individual human observers (96% on average). Most discriminative models are texture biased instead.

---
\*Equal contribution

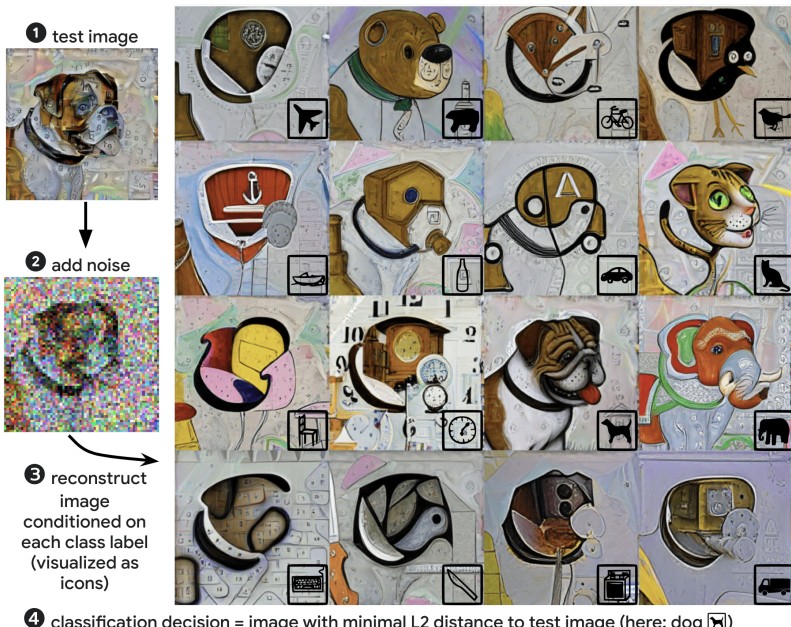

Figure 2: **Classification with a diffusion generative classifier.** Given a test image, such as a dog with clock texture (1), a text-to-image generative classifier adds random noise (2) and then reconstructs the image conditioned on the prompt "A bad photo of a <class>" for each class (3). The reconstructed image closest to the test image in $L_2$ distance is taken as the classification decision (4); this estimates the diffusion variational lower bound (Clark & Jaini, 2023). For visualization purposes, icons corresponding to the prompt class are superimposed on the reconstructed images.

# 1 INTRODUCTION

Many discriminative classifiers perform well on data similar to the training distribution, but struggle on out-of-distribution images. For instance, a cow may be correctly recognized when photographed in a typical grassy landscape, but is not correctly identified when photographed on a beach (Beery et al., 2018). In contrast to many *discriminatively* trained models, *generative* text-to-image models appear to have acquired a detailed understanding of objects: they have no trouble generating cows on beaches or dog houses made of sushi (Saharia et al., 2022). This raises the question: If we could somehow get classification decisions out of a generative model, how well would it perform out-of-distribution? For instance, would it be biased towards textures like most discriminative models or towards shapes like humans (Baker et al., 2018; Geirhos et al., 2019; Wichmann & Geirhos, 2023)?

We here investigate perceptual properties of *generative classifiers*, i.e., models trained to generate images from which we extract zero-shot classification decisions. We focus on two of the most successful types of text-to-image generative models—diffusion models and autoregressive models—and compare them to both discriminative models (e.g., ConvNets, vision transformers, CLIP) and human psychophysical data. Specifically, we focus on the task of visual object recognition (also known as classification) of challenging out-of-distribution datasets and visual illusions.

On a broader level, the question of whether perceptual processes such as object recognition are best implemented through a discriminative or a generative model has been discussed in various research communities for a long time. Discriminative inference is typically described as fast yet potentially prone to shortcut learning (Geirhos et al., 2020a), while generative modeling is often described as slow yet potentially more capable of robust inference (DiCarlo et al., 2021). The human brain appears to combine the best of both worlds, achieving fast inference but also robust generalization. How this is achieved, i.e. how discriminative and generative processes may be integrated has been described as "the deep mystery in vision" (Kriegeskorte, 2015, p. 435) and seen widespread interest in Cognitive Science and Neuroscience (see DiCarlo et al., 2021, for an overview). This mystery dates back to the idea of vision as inverse inference proposed more than 150 years ago by

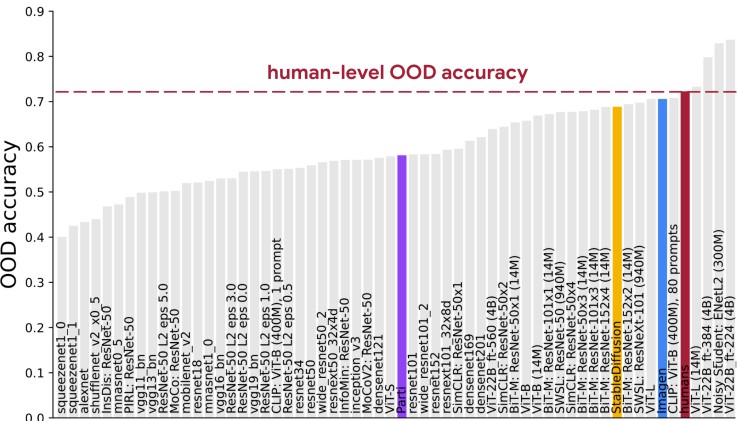

Figure 3: **Out-of-distribution accuracy** across 17 challenging datasets (Geirhos et al., 2021). Detailed results for all parametric datasets are plotted in Figure 5; Table 3 lists accuracies.

von Helmholtz (1867), who argued that the brain may need to infer the likely causes of sensory information—a process that requires a generative model of the world. In machine learning, this idea inspired approaches such as the namesake Helmholtz machine (Dayan et al., 1995), the concept of vision as Bayesian inference (Yuille & Kersten, 2006) and other analysis-by-synthesis methods (Revow et al., 1996; Bever & Poeppel, 2010; Schott et al., 2018; Zimmermann et al., 2021). However, when it comes to challenging real-world tasks like object recognition from photographs, the ideas of the past often lacked the methods (and compute power) of the future: until very recently, it was impossible to compare generative and discriminative models of object recognition simply because the only models capable of recognizing challenging images were standard discriminative models like deep convolutional networks (Krizhevsky et al., 2012; He et al., 2015) and vision transformers (Dosovitskiy et al., 2021). Excitingly, this is changing now and thus enables us to compare generative classifiers against both discriminative models and human object recognition data.

Concretely, in this work, we study the properties of generative classifiers based on three different text-to-image generative models: Stable Diffusion (SD), Imagen, and Parti on 17 challenging OOD generalization datasets from the model-vs-humans toolbox (Geirhos et al., 2021). We compare the performance of these generative classifiers with 52 discriminative models and human psychophysical data. Based on our experiments, we observe four intriguing properties of generative classifiers:

1. a human-like shape bias (Subsection 3.1),
2. near human-level out-of-distribution accuracy (Subsection 3.2),
3. state-of-the-art error consistency with humans (Subsection 3.3),
4. an understanding of certain perceptual illusions (Subsection 3.4).

## 2 METHOD: GENERATIVE MODELS AS ZERO-SHOT CLASSIFIERS

We begin with a dataset, $\mathcal{D}_n := \{(\boldsymbol{x}_1, y_1), (\boldsymbol{x}_2, y_2) \cdots, (\boldsymbol{x}_n, y_n)\}$ of $n$ images where each image belongs to one of $K$ classes $[y_K] := \{y_1, \cdots, y_k\}$. Our method classifies an image by predicting the most probable class assignment assuming a uniform prior over classes:

$$\tilde{y} = \arg\max_{y_k} p(y = y_k | \boldsymbol{x}) = \arg\max_{y_k} \ p(\boldsymbol{x} | y = y_k) \cdot p(y = y_k) = \arg\max_{y_k} \ \log p(\boldsymbol{x} | y = y_k) \quad (1)$$

A generative classifier (Ng & Jordan, 2001) uses a conditional generative model to estimate the likelihood $p_\theta(\boldsymbol{x} | y = y_k)$ where $\theta$ are the model parameters.

**Generative models:** We study the properties of three different text-to-image generative models namely Imagen (Saharia et al., 2022) which is a pixel space based diffusion model, Stable Diffusion (SD) (Rombach et al., 2022) which is a latent space based diffusion model, and Parti (Yu et al., 2022) which is a sequence-to-sequence based autoregressive model. Since these models are conditioned

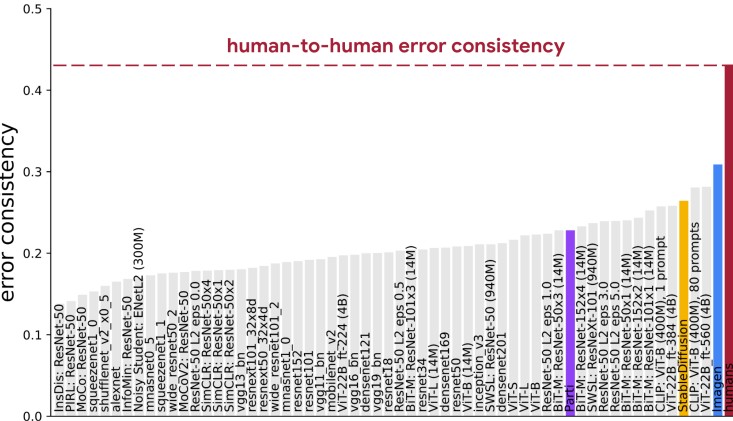

Figure 4: **Error consistency** across 17 challenging datasets (Geirhos et al., 2021). This metric measures whether errors made by models align with errors made by humans (higher is better).

on text prompts rather than class labels, we modify each label, $y_k$, to a text prompt using the template $y_k \rightarrow$ A bad photo of a $y_k$. to generate classification decisions. Conceptually, our approach to obtain classification decisions is visualized in Figure 2.

Following Clark & Jaini (2023), we generate classification decisions from diffusion models like Stable Diffusion and Imagen by approximating the conditional log-likelihood $\log p_\theta(\boldsymbol{x}|y = y_k)$ using the diffusion variational lower bound (see Appendix A for a background on diffusion models):

$$\tilde{y} = \underset{y_k}{\arg\max} \ \log p_\theta(\boldsymbol{x}|y = y_k) \approx \underset{y_k}{\arg\min} \ \mathbb{E}_{\epsilon,t}\left[\boldsymbol{w}_t \|\boldsymbol{x} - \tilde{\boldsymbol{x}}_\theta(\boldsymbol{x}_t, y_k, t)\|_2^2\right] \quad (2)$$

For SD, $\boldsymbol{x}$ is a latent representations whereas for Imagen $\boldsymbol{x}$ consists of raw image pixels.

Evaluating $p_\theta(\boldsymbol{x}|y = y_k)$ for Parti amounts to performing one forward pass of the model since it is an autoregressive model that provides an exact conditional likelihood. Thus, for each of these models we evaluate the conditional likelihood, $p_\theta(\boldsymbol{x}|y = y_k)$, for each class $y_k \in [y_K]$ and assign the class with the highest likelihood obtained via Equation (1).

**Model-vs-human datasets:** We study the performance of these generative classifiers on 17 challenging out-of-distribution (OOD) datasets proposed in the model-vs-human toolbox (Geirhos et al., 2021). Of these 17 datasets, five correspond to a non-parametric single manipulation (sketches, edge-filtered images, silhouettes, images with a texture-shape cue conflict, and stylized images where the original image texture is replaced by the style of a painting). The other twelve datasets consist of parametric image distortions like low-pass filtered images, additive uniform noise, etc. These datasets are designed to test OOD generalization for diverse models in comparison to human object recognition performance. The human data consists of 90 human observers with a total of 85,120 trials collected in a dedicated psychophysical laboratory on a carefully calibrated screen (see Geirhos et al., 2021, for details). This allows us to compare classification data for zero-shot generative models, discriminative models and human observers in a comprehensive, unified setting.

**Preprocessing:** We preprocess the 17 datasets in the model-vs-human toolbox by resizing the images to $64 \times 64$ resolution for Imagen, $256 \times 256$ for Parti, and $512 \times 512$ for SD since these are the resolutions for the each of the base models respectively. We use the prompt, A bad photo of a $y_k$, for each dataset and every model. Although Imagen (Saharia et al., 2022) is a cascaded diffusion model consisting of a $64 \times 64$ low-resolution model and two super-resolution models, we only use the $64 \times 64$ base model for our experiments here. We use v1.4 of SD (Rombach et al., 2022) for our experiments that uses a pre-trained text encoder from CLIP to encode text and a pre-trained VAE to map images to a latent space. Finally, we use the Parti-3B model (Yu et al., 2022) consisting of an image tokenizer and an encoder-decoder transformer model that converts text-to-image generation to a sequence-to-sequence modeling problem.

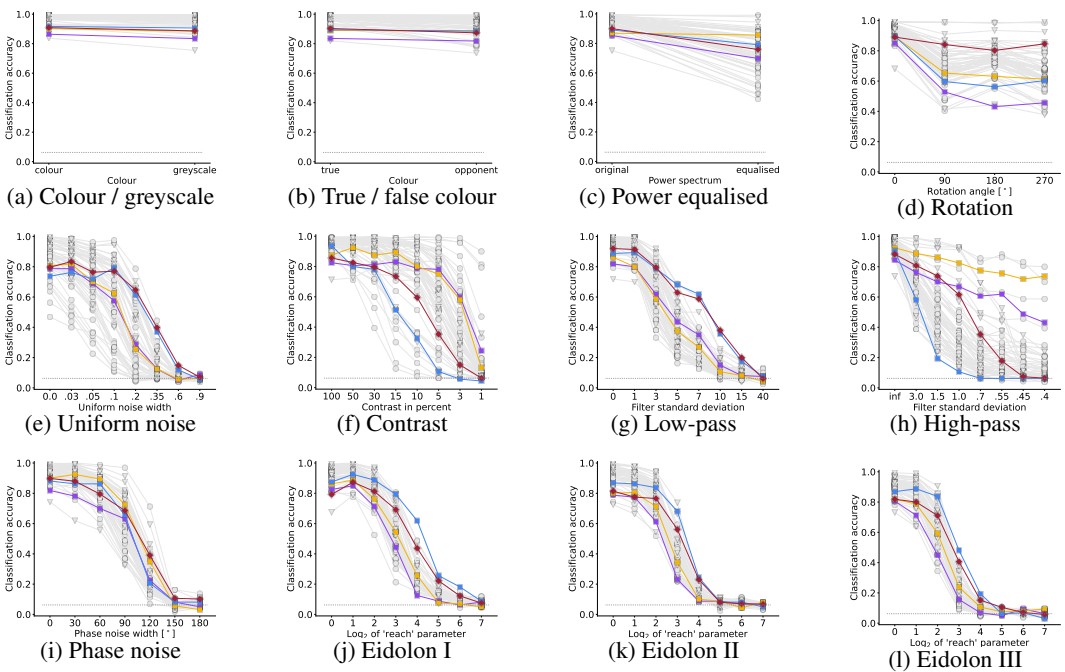

Figure 5: **Detailed out-of-distribution accuracy** for Imagen, Stable Diffusion and Parti in comparison to human observers. While not always aligning perfectly with human accuracy, the overall robustness achieved by Imagen and Stable Diffusion is comparable to that of human observers even though these models are zero-shot, i.e. neither designed nor trained to do classification.

**Baseline models for comparison:**  As baseline discriminative classifiers, we compare Imagen, SD, and Parti against 52 diverse models from the model-vs-human toolbox (Geirhos et al., 2021) that are either trained or fine-tuned on ImageNet, three ViT-22B variants (Dehghani et al., 2023) (very large 22B parameter vision transformers) and CLIP (Radford et al., 2021) as a zero-shot classifier baseline. The CLIP model is based on the largest version, ViT-L/14@224px, and consist of vision and text transformers trained with contrastive learning. We use the CLIP model that uses an ensemble of 80 different prompts for classification (Radford et al., 2021). We plot all baseline discriminative models in grey and human subject data in red.

**Metrics:**  We compare all the models over the 17 OOD datasets based on three metrics: (a) shape bias, (b) OOD accuracy and, (c) error consistency. Shape bias is defined by Geirhos et al. (2019) as the fraction of decisions that are identical to the shape label of an image divided by the fraction of decisions for which the model output was identical to either the shape or the texture label on a dataset with texture-shape cue conflict. OOD accuracy is defined as the fraction of correct decisions for a dataset that is not from the training distribution. Error consistency (see Geirhos et al., 2020b, for details) is measured in Cohen's kappa (Cohen, 1960) and indicates whether two decision makers (e.g., a model and a human observer) systematically make errors on the same images. If that is the case, it may be an indication of deeper underlying similarities in terms of how they process images and recognize objects. Error consistency between models $f_1$ and $f_2$ is defined over a dataset on which both models are evaluated on exactly the same images and output a label prediction; the metric indicates the fraction of images on which $\mathbb{1}_{f_1(x)=y_x}$ is identical to $\mathbb{1}_{f_2(x)=y_x}$ (i.e., both models are either correct or wrong on the same image) when corrected for chance agreement. This ensures that an error consistency value of 0 corresponds to chance agreement, positive values indicate beyond-chance agreement (up to 1.0) and negative values indicate systematic disagreement (down to -1.0).

| model | model type | shape bias | OOD accuracy | error consist. |
|---|---|---|---|---|
| Imagen (1 prompt) | zero-shot | **99%** | **0.71** | **0.31** |
| StableDiffuson (1 prompt) | zero-shot | 93% | 0.69 | 0.26 |
| Parti (1 prompt) | zero-shot | 92% | 0.58 | 0.23 |
| CLIP (1 prompt) | zero-shot | 80% | 0.55 | 0.26 |
| CLIP (80 prompts) | zero-shot | 57% | **0.71** | 0.28 |
| ViT-22B-384 trained on 4B images | discriminative | **87%** | **0.80** | **0.26** |
| ViT-L trained on IN-21K | discriminative | 42% | 0.73 | 0.21 |
| RN-50 trained on IN-1K | discriminative | 21% | 0.56 | 0.21 |
| RN-50 trained w/ diffusion noise | discriminative | 57% | 0.57 | 0.24 |
| RN-50 train+eval w/ diffusion noise | discriminative | 78% | 0.43 | 0.18 |

Table 1: **Benchmark results** for model-vs-human metrics (Geirhos et al., 2021). Within each model type (zero-shot vs. discriminative), the best result for each category is shown in bold.

# 3 RESULTS: FOUR INTRIGUING PROPERTIES OF GENERATIVE CLASSIFIERS

## 3.1 HUMAN-LIKE SHAPE BIAS

Introduced by Geirhos et al. (2019), the *shape bias* of a model indicates to which degree the model's decisions are based on object shape, as opposed to object texture. We study this phenomenon using the cue-conflict dataset which consists of images with shape-texture cue conflict. As shown in Geirhos et al. (2021), most discriminative models are biased towards texture whereas humans are biased towards shape (96% shape bias on average; 92% to 99% for individual observers). Interestingly, we find that all three zero-shot generative classifiers show a human-level shape bias: Imagen achieves a stunning 99% shape bias, Stable Diffusion 93% and Parti a 92% shape bias.

As we show in Figure 1, Imagen closely matches or even exceeds human shape bias across nearly all categories, achieving a previously unseeen shape bias of 99%. In Table 1, we report that all three generative classifiers significantly outperform ViT-22B (Dehghani et al., 2023), the previous state-of-the-art method in terms of shape bias, even though all three models are smaller in size, trained on less data, and unlike ViT-22B were not designed for classification.

## 3.2 NEAR HUMAN-LEVEL OOD ACCURACY

Humans excel at recognizing objects even if they are heavily distorted. *Do generative classifiers also possess similar out-of-distribution robustness?* We find that Imagen and Stable Diffusion achieve an overall accuracy that is close to human-level robustness (cf. Figure 3) despite being zero-shot models; these generative models are outperformed by some very competitive discriminative models like ViT-22B achieving super-human accuracies. The detailed plots in Figure 5 show that on most datasets (except rotation and high-pass), the performance of all three generative classifiers approximately matches human responses. Additional results are in Table 3 and Figure 11 and 12 in the appendix. Notably, all three models are considerably worse than humans in recognizing rotated images. Curiously, these models also struggle to generate rotated images when prompted with the text "A rotated image of a dog." / "An upside − down image of a dog." etc. This highlights an exciting possibility: evaluating generative models on downstream tasks like OOD datasets may be a quantitative way of gaining insights into the generation capabilities and limitations of these models.

On high-pass filtered images, Imagen performs much worse than humans whereas SD and Parti exhibit more robust performance. The difference in performance of Imagen and SD may be attributed to the weighting function used in Equation (2). Our choice of weighting function, $\boldsymbol{w}_t := \exp(-7t)$, as used in Clark & Jaini (2023) tends to give higher weight to the lower noise levels and is thus bad at extracting decisions for high-frequency images. SD on the other hand operates in the latent space and thus the weighting function in Equation (2) effects its decisions differently than Imagen. Nevertheless, this indicates that even though Imagen and SD are diffusion-based models, they exhibit very different sensitivities to high spatial frequencies. Despite those two datasets where generative clas-

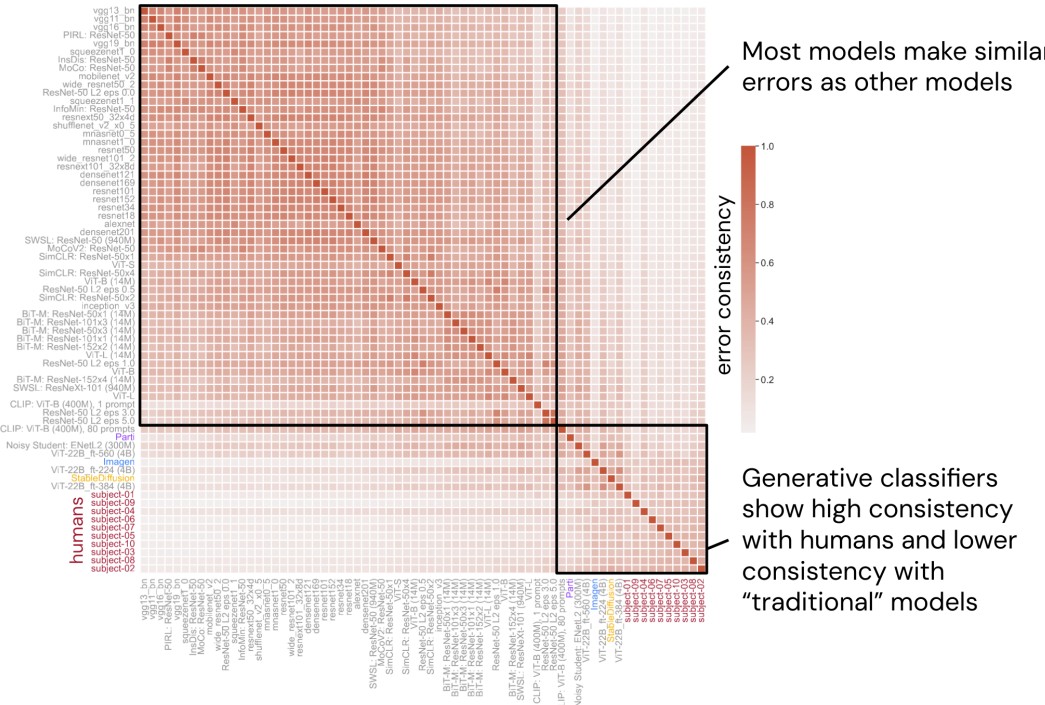

Most models make similar errors as other models

Generative classifiers show high consistency with humans and lower consistency with "traditional" models

Figure 6: **Model-to-model error consistency** for 'cue conflict' images. The matrix is sorted according to mean error consistency with humans (higher consistency from left to right / top to bottom).

sifiers show varied performance, they overall achieve impressive zero-shot classification accuracy (near human-level performance as shown in Figure 3).

### 3.3 SOTA ERROR CONSISTENCY WITH HUMAN OBSERVERS

Humans and models may both achieve, say, 90% accuracy on a dataset but do they make errors on the same 10% of images, or on different images? This is measured by *error consistency* (Geirhos et al., 2020b). In Figure 4, we show the overall results for all models across the 17 datasets. While a substantial gap towards human-to-human error consistency remains, Imagen shows the most human-aligned error patterns, surpassing previous state-of-the-art (SOTA) set by ViT-22B, a large vision transformer (Dehghani et al., 2023). SD also exhibits error consistency closer to humans but trails significantly behind Imagen. These findings appear consistent with the MNIST results by Golan et al. (2020) reporting that a generative model captures human responses better than discriminative models.

Additionally, a matrix plot of error consistency of all the models on cue-conflict images is shown in Figure 6. Interestingly, the plot shows a clear dichotomy between discriminative models that exhibit error patterns similar to each other, and generative models whose error patterns more closely match humans, thus they end up in the human cluster. While overall a substantial gap between the best models and human-to-human consistency remains (Figure 4), Imagen best captures human classification errors despite never being trained for classification. We report more detailed results in the appendix in Table 2 and Figures 11-18.

### 3.4 UNDERSTANDING CERTAIN VISUAL ILLUSIONS

Beyond quantitative benchmarking, we investigated a more qualitative aspect of generative models: whether they can understand certain visual illusions. In human perception, illusions often reveal aspects of our perceptual abilities that would otherwise go unnoticed. We therefore tested generative models on images that are visual illusions for humans. In contrast to discriminative models, generative classifiers offer a straightforward way to test illusions: for bistable images such as the famous

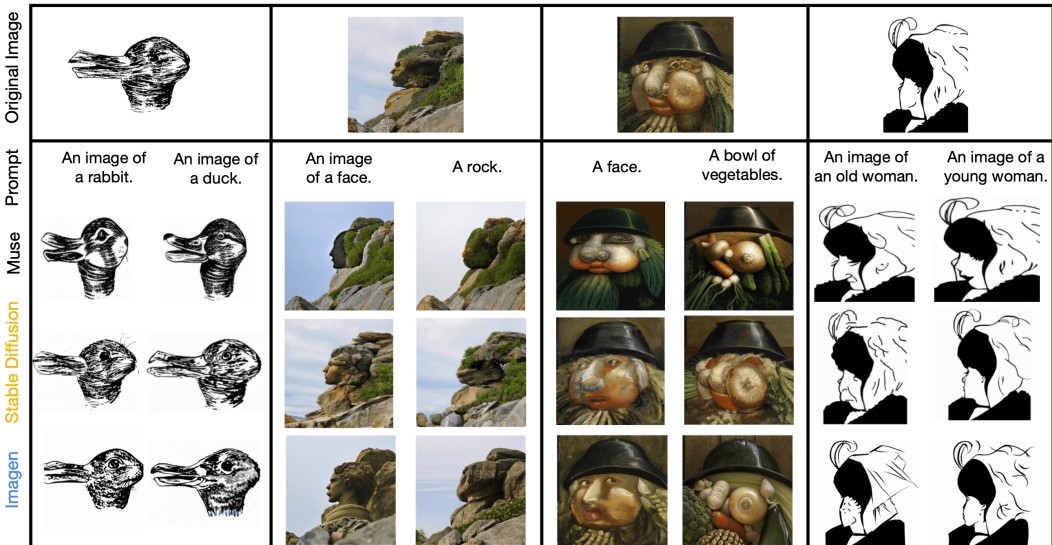

Figure 7: **Generative classifiers understand certain visual illusions** as indicated by their ability to reconstruct ambiguous images in a way that aligns with how humans perceive those images. For instance, they reconstruct a right-facing rabbit vs. a left-facing duck in the case of the bistable rabbit-duck illusion and place the face in the right location and pose for an image where humans show pareidolia (seeing patterns in things, like a face in a rock). Image attribution: Appendix D

rabbit-duck, we can prompt them to reconstruct based on 'an image of a duck' and 'an image of a rabbit'. If they can (a) reconstruct images resembling the respective animal and (b) they place the reconstructed animal in the same location and pose as humans would, this can be seen as evidence that they "understand" the illusion. We find that this is indeed the case for generative models like Imagen, Stable Diffusion, and Muse (Chang et al., 2023). Since Parti cannot directly be used for image editing, we used Muse—a masking-based generative model that operates on VQ-VAE tokens similar to Parti—for this analysis instead of Parti; Muse on the other hand cannot be used as a classifier as explained in Appendix B. This ensures that our conclusions are not limited to diffusion models but cover generative models more broadly.

In Figure 7, we use four different images that are either bistable illusions for humans or images for which humans exhibit pareidolia (a tendency to see patterns in things, like a face in a rock). In all cases, the text-to-image generative models are able to recognize the illusion and recreate correct images conditioned on the respective text prompts. This indicates that these generative models share certain bistable illusions and pareidolia with human visual perception.

# 4    ANALYSIS: WHERE DOES THE INCREASED SHAPE BIAS ORIGINATE FROM?

In Section 3, we highlighted four *intriguing properties* of generative classifiers. The most striking emergent property amongst the four is the human-level shape bias demonstrated by these generative classifiers; a bias that no discriminative models so far was able to show. A natural question to ask is thus: *What aspect of these generative models causes such an increase in shape bias?*

We observed that for diffusion models like Imagen and Stable Diffusion, the recreated images used for classification were usually devoid of texture cues (for example see Figure 2). We posit that the denoising process used for classification (cf. Equation (2)) of the diffusion model might bias it towards capturing low-frequency information and thereby focuses on the global structure of the image as captured by the shape of an object. Indeed, in Figure 5, we observe that while generative classifiers are well within the range of other models for most datasets, they demonstrate very distinctive results on low-pass filtered images (also known as blurred); Imagen—the most shape-biased model—is on par with humans. Conversely, Imagen struggles to classify high-pass images. Could it be the case that these generative models put more emphasis on lower spatial frequencies whereas most textures are high frequency in nature?

If this is indeed the case, then performance on blurred images and shape bias should have a significant positive correlation. We tested this hypothesis empirically and indeed found a strong positive and highly significant correlation between the two (Pearson's $r(58) = .59, p < 0.001$; Spearman's $r(58) = .64, p < 0.001$); a finding consistent with Subramanian et al. (2023) observing a significant correlation between shape bias and spatial frequency channel bandwidth. While correlations establish a connection between the two, they are of course not evidence for a causal link. We hypothesized that the noise applied during diffusion training might encourage models to ignore high-frequency textures and focus on shapes. To test this prediction, we trained a standard ResNet-50 on ImageNet-1K (Russakovsky et al., 2015) by adding diffusion-style noise as a data augmentation during both training and evaluation. Interestingly, such a model trained with data augmented with diffusion style noise causes an increase in shape bias from 21% for a standard ResNet-50 to 78% as shown in Figures 1 and 14 and Table 1. This simple trick achieves a substantially higher shape bias than the 62% observed by prior work when combining six different techniques and augmentations (Hermann et al., 2020).

This result shows that (i) diffusion style training biases the models to emphasize low spatial frequency information and (ii) models that put emphasis on lower spatial frequency noise exhibit increased shape bias. Other factors such as generative training, the quality and quantity of data, and the use of a powerful language model might also play a role. However, given the magnitude of the observed change in shape bias this indicates that diffusion-style training is indeed a crucial factor.

## 5 DISCUSSION

**Motivation.** While generative pre-training has been prevalent in natural language processing, in computer vision it is still common to pre-train models on labeled datasets such as ImageNet (Deng et al., 2009) or JFT (Sun et al., 2017). At the same time, generative text-to-image models like Stable Diffusion, Imagen, and Parti show powerful abilities to generate photo-realistic images from diverse text prompts. This suggests that these models learn useful representations of the visual world, but so far it has been unclear how their representations compare to discriminative models. Furthermore, discriminative models have similarly dominated computational modeling of human visual perception, even though the use of generative models by human brains has long been hypothesized and discussed. In this work, we performed an empirical investigation on out-of-distribution datasets to assess whether discriminative or generative models better fit human object recognition data.

**Key results.** We report four intriguing human-like properties of *generative* models: (1) Generative classifiers are the first models that achieve a human-like shape bias (92–99%); (2) they achieve near human-level OOD accuracy despite being zero-shot classifiers that were neither trained nor designed for classification; (3) one of them (Imagen) shows the most human-aligned error patterns that machine learning models have achieved to date; and (4) all investigated models qualitatively capture the ambiguities of images that are perceptual illusions for humans.

**Implications for human perception.** Our results establish generative classifiers as one of the leading behavioral models of human object recognition. While we certainly don't resolve the "deep mystery of vision" (Kriegeskorte, 2015, p. 435) in terms of how brains might combine generative and discriminative models, our work paves the way for future studies that might combine the two. Quoting Luo (2022, p. 22) on diffusion, "It is unlikely that this is how we, as humans, naturally model and generate data; we do not seem to generate novel samples as random noise that we iteratively denoise."—we fully agree, but diffusion may just be one of many implementational ways to arrive at a representation that allows for powerful generative modeling. Human brains are likely to use a different *implementation*, but they still may (or may not) end up with a similar *representation*.

**Implications for machine perception.** We provide evidence for the benefits of generative pre-training, particularly in terms of zero-shot performance on challenging out-of-distribution tasks. In line with recent work on using generative models for depth estimation (Zhao et al., 2023) or segmentation (Burgert et al., 2022; Brempong et al., 2022; Li et al., 2023), this makes the case for generative pre-training as a compelling alternative to contrastive or discriminative training for vision tasks. Additionally, our experiments provide a framework to find potential bugs of generative models through classification tasks. For example, all the generative models performed poorly on the rotation dataset; those models also struggled to generate "rotated" or "upside-down" images of objects. Similar experiments could be used to evaluate generative models for undesirable behaviour, toxicity and bias.

**Limitations.** A limitation of the approach we used in the paper is the computational speed (as we also alluded to in Section 1). The approach does not yield a practical classifier. Secondly, all three models have different model sizes, input resolutions, and are trained on different datasets for different amounts of time, so the comparison is not perfect. Furthermore, different models use different language encoders which may be a confounding factor. Through including diverse generative models, our comparisons aim to highlight the strengths and weaknesses of generative models. We explore a number of ablations and additional analyses in the appendix, including shape bias results for image captioning models that are of generative nature, but not trained for text-to-image modeling.

**Future directions.** Beyond the questions regarding how biological brains might combine generative and discriminative models, we believe it will be interesting to study how, and to what degree, language cross-attention influences the intriguing properties we find. Moreover, is denoising diffusion training a crucial component that explains the impressive performance of Imagen and SD? We hope our findings show how intriguing generative classifiers are for exploring exciting future directions.

ACKNOWLEDGMENTS

We would like to express our gratitude to the following colleagues (in alphabetical order) for helpful discussions and feedback: David Fleet, Katherine Hermann, Been Kim, Alex Ku, Jon Shlens, and Kevin Swersky. Furthermore, we would like to thank Michael Tschannen and Manoj Kumar for suggesting the CapPa model analysis and providing the corresponding models.

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

# Appendix

## Table of Contents

## A  BACKGROUND ON DIFFUSION MODELS

Diffusion models (Sohl-Dickstein et al., 2015; Ho et al., 2020; Song et al., 2020; Song & Ermon, 2020) are latent variable generative models defined by a forward and reverse Markov chain. Given an unknown data distribution, $q(\boldsymbol{x}_0)$, over observations, $\boldsymbol{x}_0 \in \mathbb{R}^d$, the forward process corrupts the data into a sequence of noisy latent variables, $\boldsymbol{x}_{1:T} := \{\boldsymbol{x}_1, \boldsymbol{x}_2, \cdots, \boldsymbol{x}_T\}$, by gradually adding Gaussian noise with a fixed schedule defined as:

$$q(\boldsymbol{x}_{1:T}|\boldsymbol{x}_0) := \prod_{t=1}^{T} q(\boldsymbol{x}_t|\boldsymbol{x}_{t-1}) \tag{3}$$

where $q(\boldsymbol{x}_t|\boldsymbol{x}_{t-1}) := \mathsf{Normal}(\boldsymbol{x}_t; \sqrt{1-\beta_t}\boldsymbol{x}_{t-1}, \beta_t\boldsymbol{I})$. The reverse Markov process gradually denoises the latent variables to the data distribution with learned Gaussian transitions starting from $\mathsf{Normal}(\boldsymbol{x}_T; 0, \boldsymbol{I})$ i.e.

$$p_{\boldsymbol{\theta}}(\boldsymbol{x}_{0:T}) := p(\boldsymbol{x}_T) \cdot \prod_{t=0}^{T-1} p_{\boldsymbol{\theta}}(\boldsymbol{x}_{t-1}|\boldsymbol{x}_t)$$

$p_{\boldsymbol{\theta}}(\boldsymbol{x}_{t-1}|\boldsymbol{x}_t) := \mathsf{Normal}(\boldsymbol{x}_{t-1}; \boldsymbol{\mu}_{\boldsymbol{\theta}}(\boldsymbol{x}_t, t), \boldsymbol{\Sigma}_{\boldsymbol{\theta}}(\boldsymbol{x}_t, t))$. The aim of training is for the forward process distribution $\{\boldsymbol{x}_t\}_{t=0}^T$ to match that of the reverse process $\{\tilde{\boldsymbol{x}}_t\}_{t=0}^T$ i.e., the generative model $p_{\boldsymbol{\theta}}(\boldsymbol{x}_0)$ closely matches the data distribution $q(\boldsymbol{x}_0)$. Specifically, these models can be trained by optimizing the variational lower bound of the marginal likelihood (Ho et al., 2020; Kingma et al., 2021):

$$-\log p_{\theta}(\boldsymbol{x}_0) \leq -\mathsf{VLB}(\boldsymbol{x}_0) := \mathcal{L}_{\mathsf{Prior}} + \mathcal{L}_{\mathsf{Recon}} + \mathcal{L}_{\mathsf{Diffusion}}$$

$\mathcal{L}_{\mathsf{Prior}}$ and $\mathcal{L}_{\mathsf{Recon}}$ are the prior and reconstruction loss that can be estimated using standard techniques in the literature (Kingma & Welling, 2014). The (re-weighted) diffusion loss can be written as:

$$\mathcal{L}_{\mathsf{Diffusion}} = \mathbb{E}_{\boldsymbol{x}_0, \boldsymbol{\varepsilon}, t}\left[\boldsymbol{w}_t\|\boldsymbol{x}_0 - \tilde{\boldsymbol{x}}_{\boldsymbol{\theta}}(\boldsymbol{x}_t, t)\|_2^2\right]$$

with $\boldsymbol{x}_0 \sim q(\boldsymbol{x}_0)$, $\boldsymbol{\varepsilon} \sim \mathsf{Normal}(0, \boldsymbol{I})$, and $t \sim \mathcal{U}([0, T])$. Here, $\boldsymbol{w}_t$ is a weight assigned to the timestep, and $\tilde{\boldsymbol{x}}_{\boldsymbol{\theta}}(\boldsymbol{x}_t, t)$ is the model's prediction of the observation $\boldsymbol{x}_0$ from the noised observation $\boldsymbol{x}_t$. Diffusion models can be conditioned on additional inputs like class labels, text prompts, segmentation masks or low-resolution images, in which case $\tilde{\boldsymbol{x}}_{\boldsymbol{\theta}}$ also takes a conditioning signal $\boldsymbol{y}$ as input.

**Design choices zero-shot classification using diffusion models:** We follow the exact experiment setting here as in Clark & Jaini (2023) for Imagen and Stable Diffusion to obtain classification decisions. Specifically, we use the heuristic weighting function $w_t := \exp(-7t)$ in Equation (2) to aggregate scores across multiple time steps. We use a single prompt for each image instead of an ensemble of prompts as used in CLIP to keep the experiments simple.

**Loss function:** We use the $L_2$ loss function for diffusion-based models since it approximates the diffusion variational lower bound (see Equation (2)) and thus results in a Bayesian classifier. Furthermore, both Stable Diffusion and Imagen are trained with the $L_2$ loss objective. Thus, a different loss function will no longer result in a Bayesian classifier and will not work well due to differences from the training paradigm.

---

**Algorithm 1** Classification using diffusion models.

---

**given**: Example to classify $x$, diffusion model w/ params $\theta$, weighting function $w$.

*Map from classes to diffusion model scores.*
scores $= \{y_i : [] \text{ for } y_i \in [y_K]\}$
$n = 0$
$n < $ max_scores:
$\quad n = n + 1$
$\quad$ *Noise the image*
$\quad t \sim \mathcal{U}([0, 1])$
$\quad x_t \sim q(x_t | x)$
$\quad$ *Score against the remaining classes.*
$\quad$ **for** $y_i \in$ scores:
$\quad\quad$ add $w_t \| x - \tilde{x}_{\theta}(x_t, \phi(y_i), t) \|_2^2$
$\quad\quad$ to scores[$y_i$]
**return** $\tilde{y}$

---

## B   MUSE AS A CLASSIFIER

In Figure 8, we visualize why we were not able to include Muse as a (successful) classifier in our experiments. Even on clean, undistorted images Muse achieved only approximately chance-level accuracy. This may, however, just be a limitation on how we attempted to extract classification decisions out of the model; it is very well possible that other approaches might work better.

## C   LIMITATIONS

As mentioned in the introduction, using a generative model comes with advantages and disadvantages: potentially better generalization currently comes at the cost of being slower computationally compared to standard discriminative models. While this doesn't matter much for the purpose of analyses, it is a big drawback in practical applications and any approaches that improve speed would be most welcome—in particular, generating at least one prediction (i.e., image generation) per class as we currently do is both expensive and slow.

Furthermore, the models we investigate all differ from another in more than one ways. For instance, their training data, architecture, and training procedure is not identical, thus any differences between the models cannot currently be attributed to a single factor. That said, through the inclusion of a set of diverse models covering pixel-based diffusion, latent space diffusion, and autoregressive models we seek to at least cover a variety of generative classifiers in order to ensure that the conclusions we draw are not limited to a narrow set of generative models.

## D   IMAGE ATTRIBUTION

**Rabbit-duck image:**
Attribution: Unknown source, Public domain, via Wikimedia Commons.

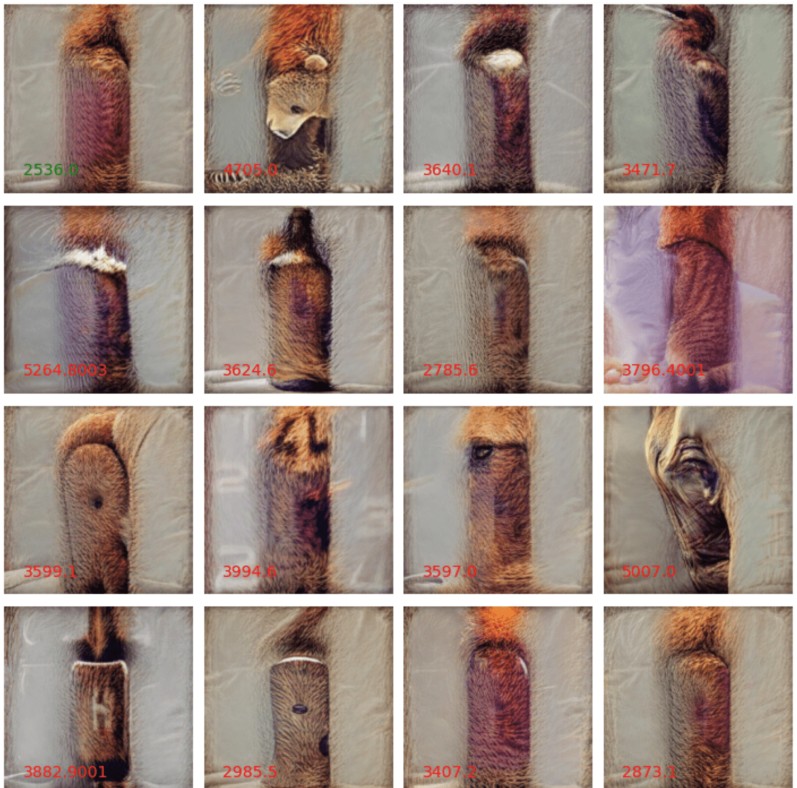

Figure 8: **Muse as a classifier.** This figure illustrates why Muse reconstructions are often challenging to use for classification. Given a starting image of a bottle with bear texture, we here plot 16 reconstructions each prompted with a different label. The $\mathcal{L}_2$ distance to the original image is shown in red for all categories except the one with the lowest distance (here: top left), for which the distance is plotted in green. The images with the lowest distance (row 1 column 1: airplane; row 2 column 3: car, row 4 column 4: truck) appear to be categories for which the model is unable to generate a realistic reconstruction, and thus simply sticks close to the original image. The image generated by prompting with the correct shape category (bottle) is shown in row 2 column 2. Since Muse returns images fairly close to the original one if it is not able to generate a realistic reconstruction, the approach of measuring $\mathcal{L}_2$ distance between original and reconstruction is not a feasible classification approach for Muse.

Link: https://upload.wikimedia.org/wikipedia/commons/9/96/Duck-Rabbit.png

**Rock image:**
Attribution: Mirabeau, CC BY-SA 3.0, via Wikimedia Commons.
Link: https://commons.wikimedia.org/wiki/File:Visage_dans_un_rocher.jpg

**Vegetable portrait image:**
Attribution: Giuseppe Arcimboldo, Public domain, via Wikimedia Commons.
Link: https://upload.wikimedia.org/wikipedia/commons/4/49/Arcimboldo_Vegetables.jpg

**Woman image:**
Attribution: W. E. Hill, Public domain, via Wikimedia Commons.
Link: https://upload.wikimedia.org/wikipedia/commons/5/5f/My_Wife_and_My_Mother-In-Law_%28Hill%29.svg

# E    DETAILS ON RESNET-50 TRAINING WITH DIFFUSION NOISE

We trained a ResNet-50 in exactly the same way as for standard, 90 epoch JAX ImageNet training with the key difference that we added diffusion noise as described by the code below. Since this makes the training task substantially more challenging, we trained the model for 300 instead of 90 epochs. The learning rate was 0.1 with a cosine learning rate schedule, 5 warmup epochs, SGD momentum of 0.9, weight decay of 0.0001, and a per device batch size of 64. For diffusion style denoising we used a flag named "sqrt_alphas" which ensures that the noise applied doesn't completely destroy the image information in most cases. The input to the AddNoise method is in the [0, 1] range; the output of the AddNoise method exceeds this bound due to the noise; we did not normalize / clip it afterwards but instead directly fed this into the network. We did not perform ImageNet mean/std normalization. The training augmentations we used were 1. random resized crop, 2. random horizontal flip, 3. add diffusion noise. We did not optimize any of those settings with respect to any of the observed findings (e.g., shape bias) since we were interested in generally applicable results.

Listing 1: Python example of how diffusion noise was added as a data augmentation technique.

```python
#Copyright 2023 DeepMind Technologies Limited.
#SPDX-License-Identifier: Apache-2.0

import dataclasses
from typing import Sequence, Tuple
import grain.tensorflow as tf_grain
import jax
import tensorflow as tf

@dataclasses.dataclass(frozen=True)
class AddNoise(tf_grain.MapTransform):
  """Adds diffusion-style noise to the image."""
  sqrt_alphas: bool = True

  def map(self, features: FlatFeatures) -> FlatFeatures:
    image = features["image"]
    alpha = tf.random.uniform([])
    if self.sqrt_alphas:
      alpha = tf.sqrt(alpha)
    std = tf.sqrt(1 - alpha * alpha)
    image = image * alpha + std * tf.random.normal(image.shape)
    features["image"] = image
    features["noise_level"] = std
    return features
```

# F    SHAPE BIAS OF IMAGE CAPTIONING MODELS

We discovered that text-to-image generative models, when turned into classifiers, have a high shape bias. We were interested in understanding whether other forms of generative modeling (beyond text-to-image) could lead to similar results. As an initial step towards exploring this question, we tested the CapPa models from Tschannen et al. (2023); those models are of generative nature too but instead of text-to-image modeling they are trained to produce image captions. The results are shown in Figure 9. Both CapPa variants (B16 and L16) are more shape biased than most discriminative models, but at the same time both are less shape biased than the text-to-image generative classifiers we tested. This could indicate that generative modeling is helpful when it comes to shape bias, while the objective function and data determine how strong the shape bias is.

**Method details for CapPa evaluation.** We obtained the original B16 and L16 checkpoints from the CapPa paper. We used the identical prompt as for the generative classifiers, A bad photo of a class where "class" is substituted for e.g. "car" (and using 'an' for classes starting

with a vowel). We only used this single prompt since initial experiments showed that CapPa ImageNet validation accuracy decreases when using the 80 CLIP ImageNet template prompts (w.r.t. just using a single prompt). This could be attributed to the fact that the CapPa models were not designed to handle different prompts well.

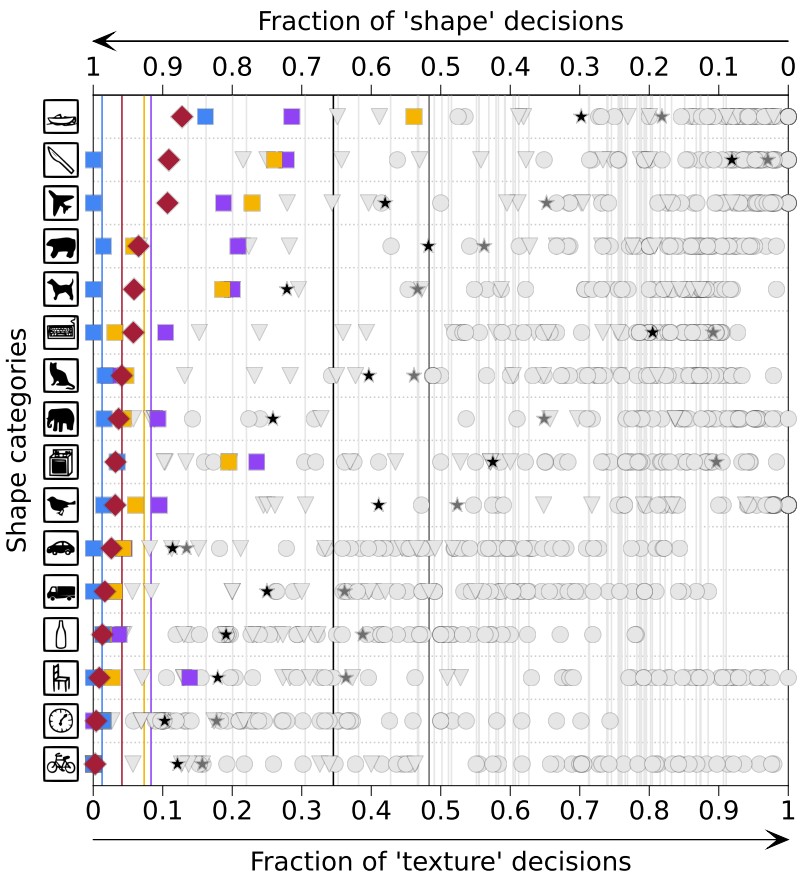

Figure 9: **Shape bias of CapPa models.** This plot is identical to Figure 1 but contains two additional datapoints: the CapPa models which are trained to produce text captions for images. The grey stars correspond to CapPa B16; the black stars correspond to CapPa L16. Both models are trained on 9B image-text pairs.

## G  ADDITIONAL PLOTS FOR MODEL-VS-HUMAN BENCHMARK

We here plot detailed performance for all models with respect to a few different properties of interest / metrics:

- Aggregated performance across 17 datasets: Figure 10
- Out-of-distribution accuracy: Figure 11 for parametric datasets and Figure 12 for nonparametric datasets
- Model-to-human error consistency: Figure 11
- Human-to-human, model-to-model error consistency (for nonparametric datasets): Figures 6 and 15 to 18
- Shape bias: Figure 13 and Figure 14

All metrics are based on the model-vs-human toolbox and explained in more detail in (Geirhos et al., 2021).

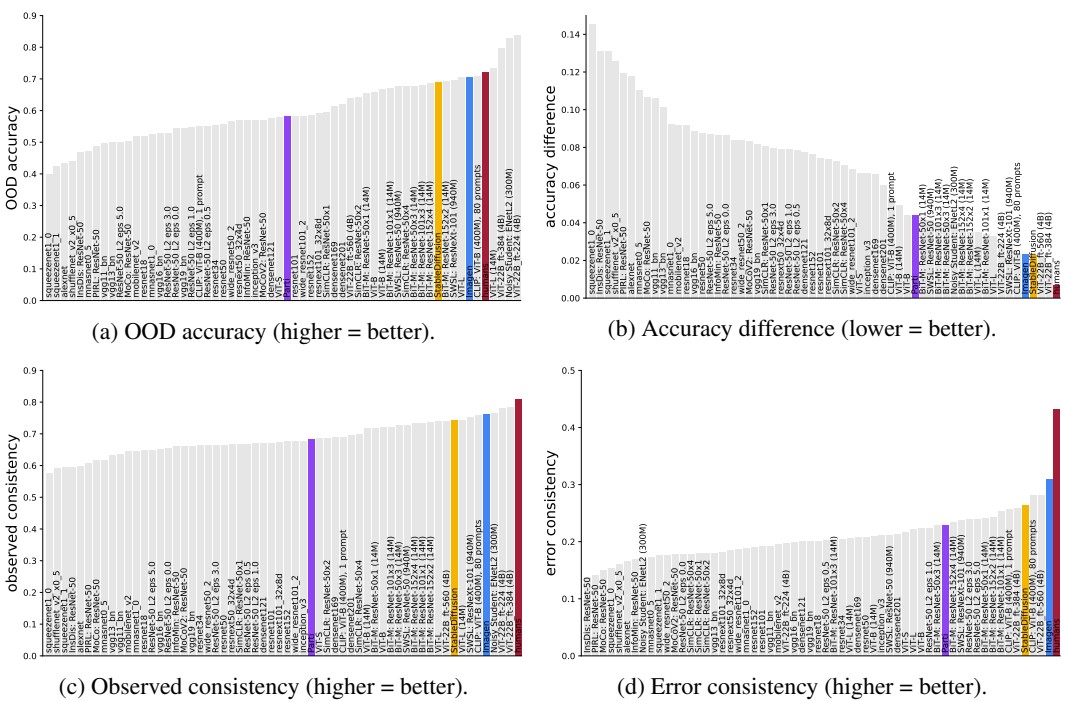

(a) OOD accuracy (higher = better).

(b) Accuracy difference (lower = better).

(c) Observed consistency (higher = better).

(d) Error consistency (higher = better).

Figure 10: Benchmark results for different models, aggregated over datasets.

# H QUANTITATIVE BENCHMARK SCORES AND RANKINGS

Table 2 and Table 3 list the detailed performance aggregated across 17 datasets for each model, with the former focusing on metrics related to "most human-like object recognition behavior" and the latter focusing on out-of-distribution accuracy.

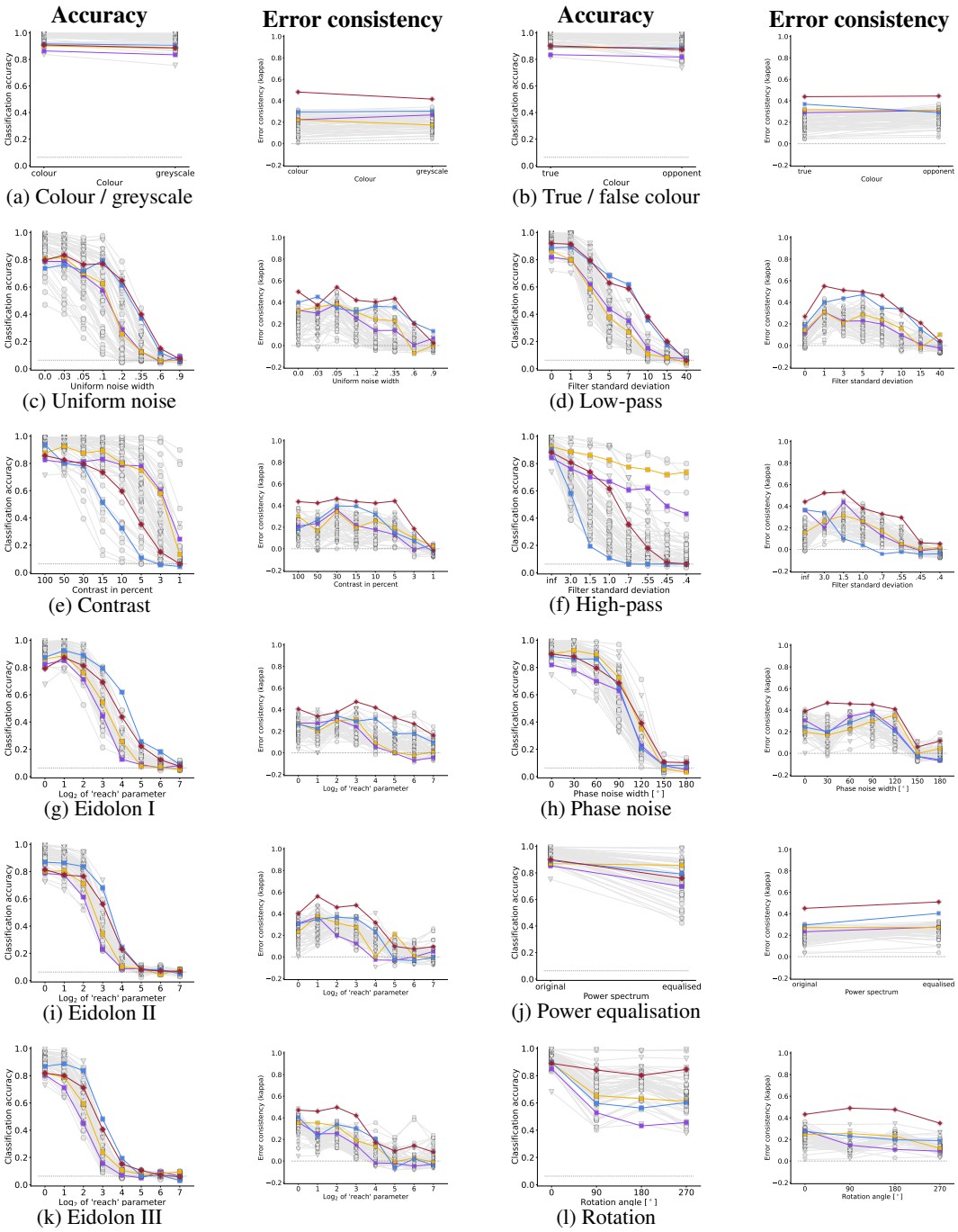

Figure 11: OOD accuracy and error consistency across all twelve parametric datasets from Geirhos et al. (2021). Error consistency results for nonparametric datasets are plotted in Figures 6 and 15 to 18.

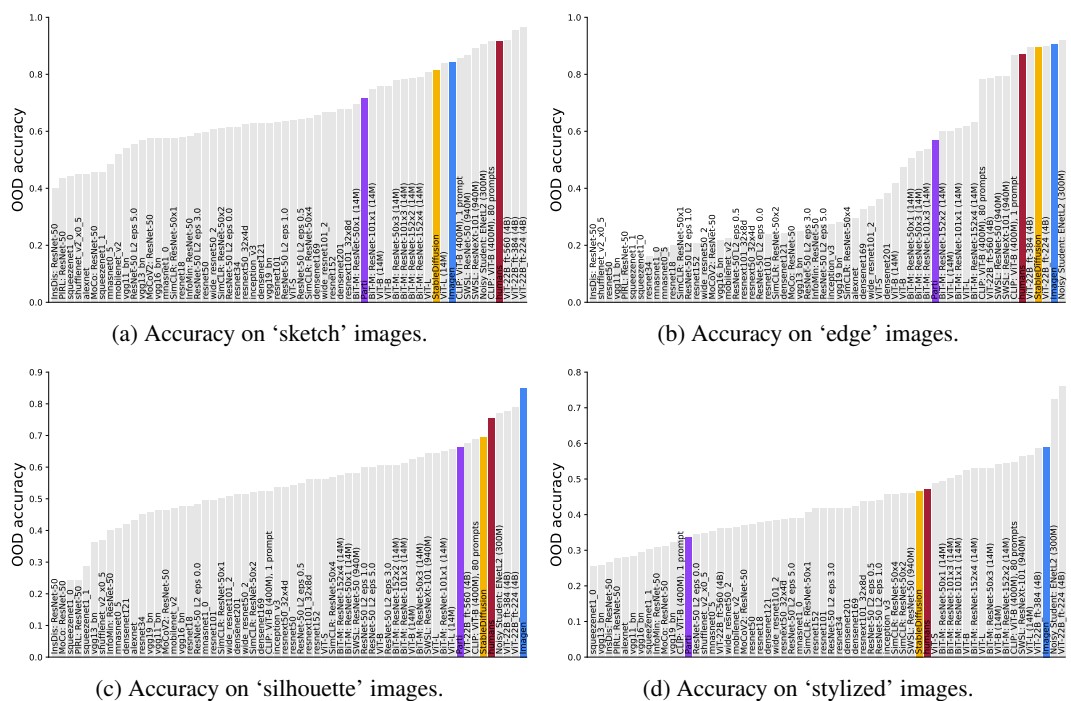

(a) Accuracy on 'sketch' images.

(b) Accuracy on 'edge' images.

(c) Accuracy on 'silhouette' images.

(d) Accuracy on 'stylized' images.

Figure 12: OOD accuracy on all four nonparametric datasets (i.e., datasets with only a single corruption type and strength).

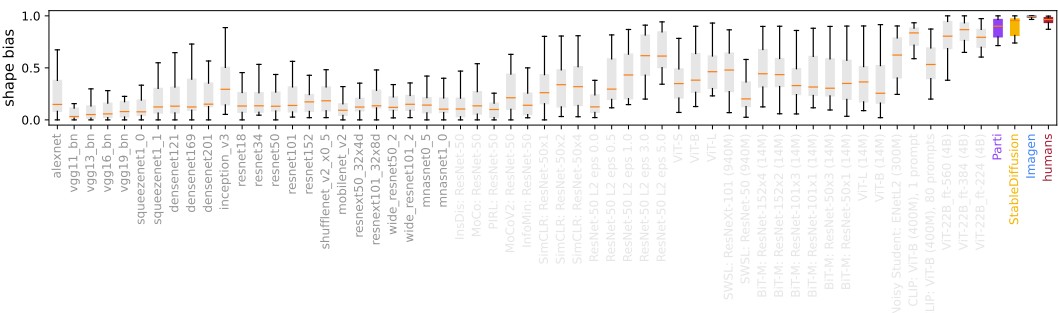

Figure 13: Zero-shot generative classifiers achieve a **human-level shape bias**: 99% for Imagen, 93% for Stable Diffusion, 92% for Parti and 92–99% for individual human observers (96% on average). This figure shows boxplots highlighting the spread across 16 categories for each model as a different way of visualizing the data from Figure 1.

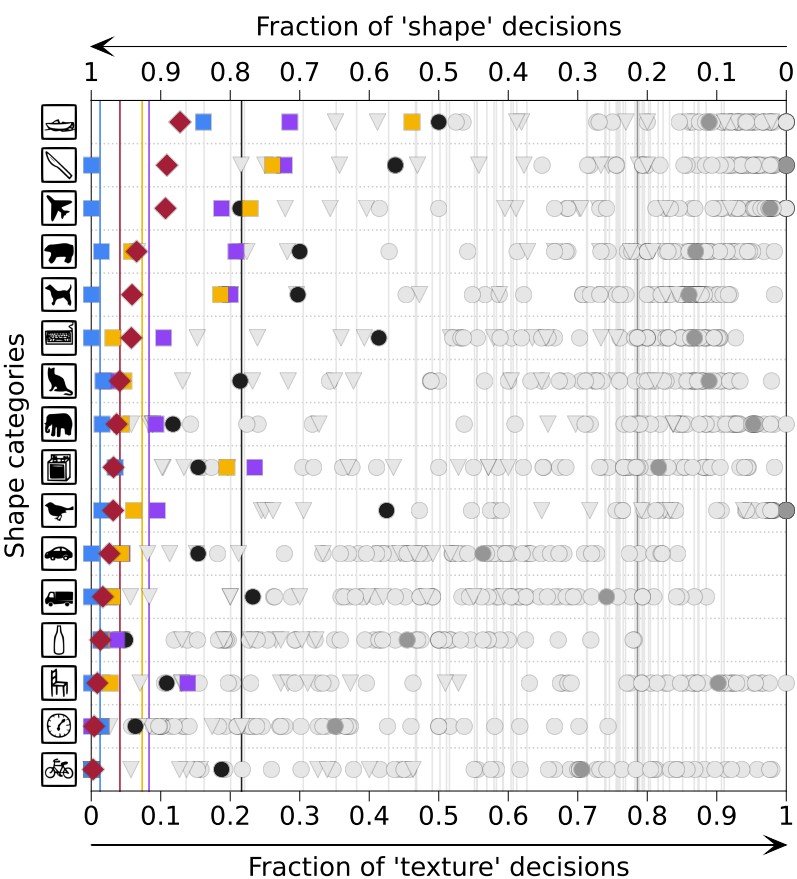

Figure 14: Shape bias before and after training with diffusion noise: This figure shows how ResNet-50 shape bias increases from 21% for a vanilla model (dark grey circles) to 78% when trained and evaluated with diffusion noise (black circles). Horizontal lines indicate the average shape bias across categories. Other models as in Figure 13.

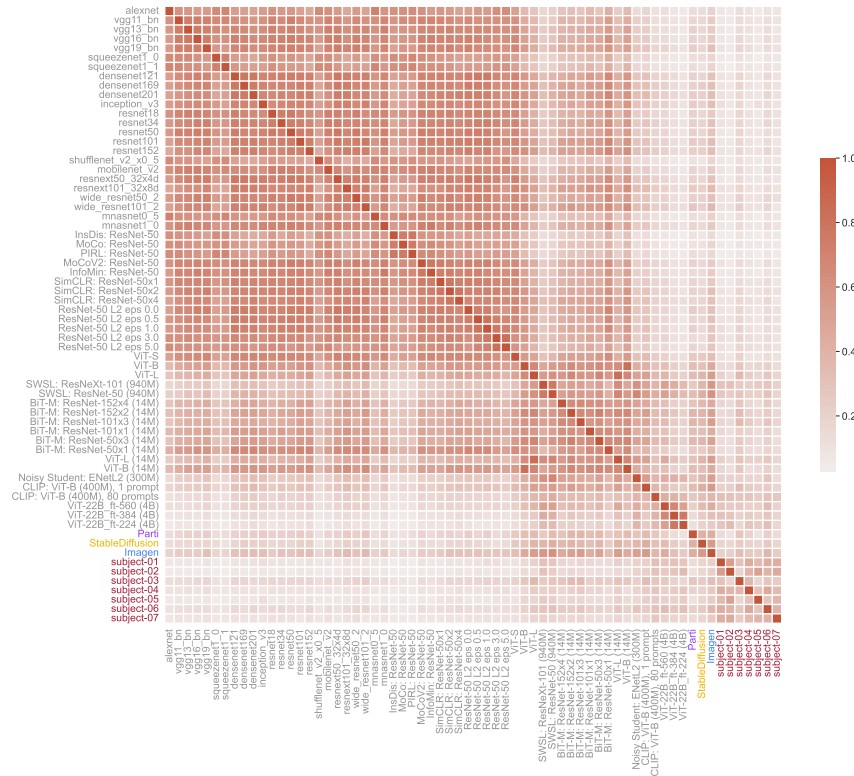

Figure 15: Error consistency for 'sketch' images.

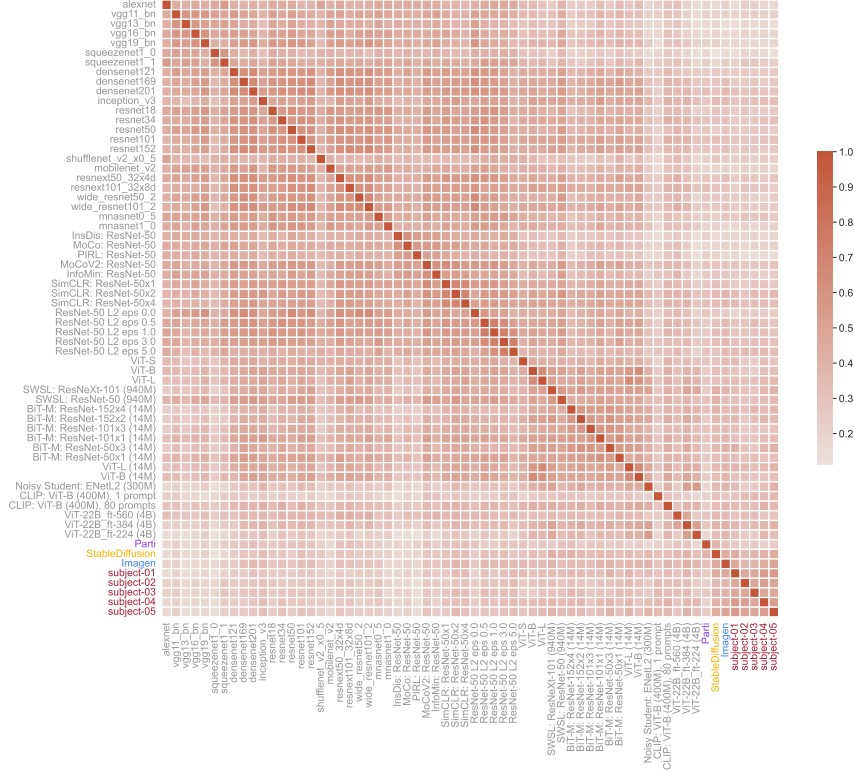

Figure 16: Error consistency for 'stylized' images.

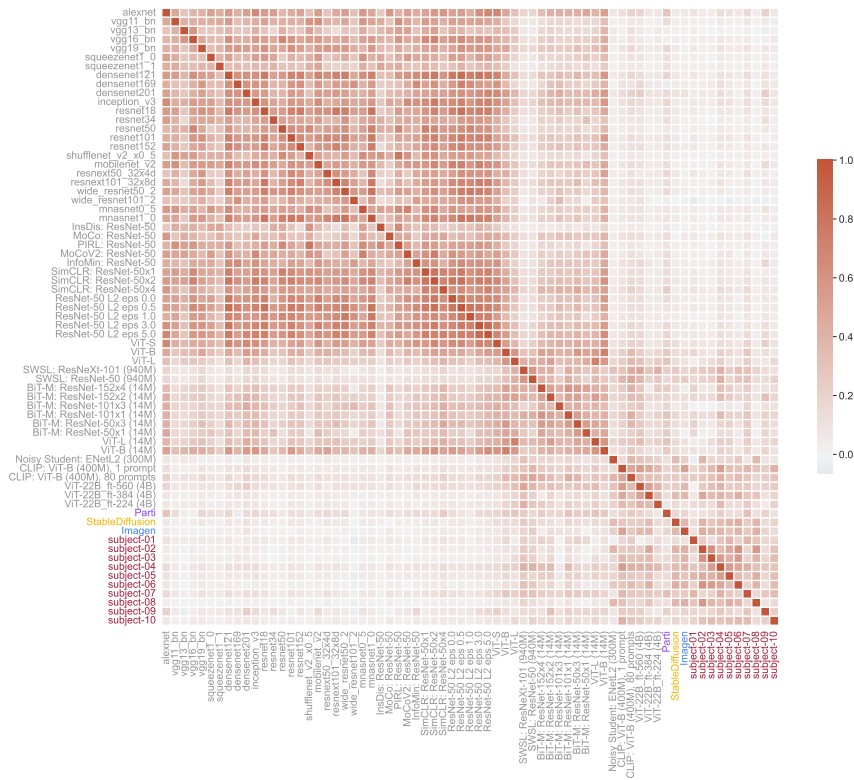

Figure 17: Error consistency for 'edge' images.

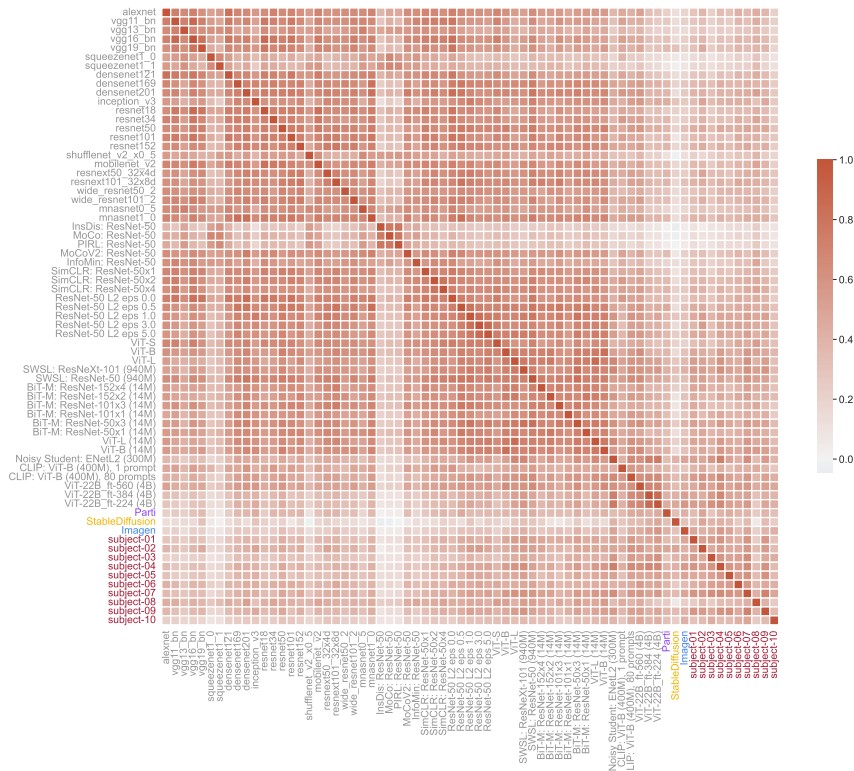

Figure 18: Error consistency for 'silhouette' images.

Table 2: Benchmark table of model results for most human-like behaviour, aggregated over all 17 datasets from Geirhos et al. (2021). The three metrics "accuracy difference" "observed consistency" and "error consistency" each produce a different model ranking. The mean rank of a model across those three metrics is used to rank the models on our benchmark.

| model | accuracy diff. ↓ | obs. consistency ↑ | error consistency ↑ | mean rank ↓ |
|---|---|---|---|---|
| ViT-22B_ft-384 (4B) | **0.018** | **0.783** | 0.258 | **2.333** |
| Imagen (860M) | 0.023 | 0.761 | **0.309** | 3.000 |
| ViT-22B_ft-560 (4B) | 0.022 | 0.739 | 0.281 | 4.333 |
| CLIP: ViT-B (400M), 80 prompts | 0.023 | 0.758 | 0.281 | 4.333 |
| StableDiffusion | 0.023 | 0.743 | 0.264 | 5.000 |
| SWSL: ResNeXt-101 (940M) | 0.028 | 0.752 | 0.237 | 8.000 |
| BiT-M: ResNet-101x1 (14M) | 0.034 | 0.733 | 0.252 | 9.333 |
| BiT-M: ResNet-152x2 (14M) | 0.035 | 0.737 | 0.243 | 10.000 |
| ViT-L | 0.033 | 0.738 | 0.222 | 11.667 |
| BiT-M: ResNet-152x4 (14M) | 0.035 | 0.732 | 0.233 | 12.667 |
| ViT-L (14M) | 0.035 | 0.744 | 0.206 | 14.000 |
| ViT-22B_ft-224 (4B) | 0.030 | 0.781 | 0.197 | 14.000 |
| BiT-M: ResNet-50x3 (14M) | 0.040 | 0.726 | 0.228 | 14.333 |
| BiT-M: ResNet-50x1 (14M) | 0.042 | 0.718 | 0.240 | 14.667 |
| CLIP: ViT-B (400M), 1 prompt | 0.054 | 0.688 | 0.257 | 16.000 |
| SWSL: ResNet-50 (940M) | 0.041 | 0.727 | 0.211 | 16.667 |
| ViT-B | 0.044 | 0.719 | 0.223 | 17.000 |
| BiT-M: ResNet-101x3 (14M) | 0.040 | 0.720 | 0.204 | 19.333 |
| ViT-B (14M) | 0.049 | 0.717 | 0.209 | 20.000 |
| densenet201 | 0.060 | 0.695 | 0.212 | 20.333 |
| Noisy Student: ENetL2 (300M) | 0.040 | 0.764 | 0.169 | 22.333 |
| ViT-S | 0.066 | 0.684 | 0.216 | 22.333 |
| densenet169 | 0.065 | 0.688 | 0.207 | 23.000 |
| inception_v3 | 0.066 | 0.677 | 0.211 | 23.333 |
| ResNet-50 L2 eps 1.0 | 0.079 | 0.669 | 0.224 | 26.667 |
| ResNet-50 L2 eps 3.0 | 0.079 | 0.663 | 0.239 | 27.667 |
| SimCLR: ResNet-50x4 | 0.071 | 0.698 | 0.179 | 30.333 |
| wide_resnet101_2 | 0.068 | 0.676 | 0.187 | 30.333 |
| ResNet-50 L2 eps 0.5 | 0.078 | 0.668 | 0.203 | 31.000 |
| densenet121 | 0.077 | 0.671 | 0.200 | 31.000 |
| SimCLR: ResNet-50x2 | 0.073 | 0.686 | 0.180 | 31.333 |
| resnet152 | 0.077 | 0.675 | 0.190 | 31.667 |
| resnet101 | 0.074 | 0.671 | 0.192 | 31.667 |
| resnext101_32x8d | 0.074 | 0.674 | 0.182 | 32.667 |
| ResNet-50 L2 eps 5.0 | 0.087 | 0.649 | 0.240 | 32.667 |
| resnet50 | 0.087 | 0.665 | 0.208 | 34.333 |
| resnet34 | 0.084 | 0.662 | 0.205 | 35.000 |
| vgg19_bn | 0.081 | 0.660 | 0.200 | 35.667 |
| resnext50_32x4d | 0.079 | 0.666 | 0.184 | 36.333 |
| SimCLR: ResNet-50x1 | 0.080 | 0.667 | 0.179 | 38.000 |
| resnet18 | 0.091 | 0.648 | 0.201 | 40.333 |
| vgg16_bn | 0.088 | 0.651 | 0.198 | 40.333 |
| wide_resnet50_2 | 0.084 | 0.663 | 0.176 | 41.667 |
| MoCoV2: ResNet-50 | 0.083 | 0.660 | 0.177 | 42.000 |
| mobilenet_v2 | 0.092 | 0.645 | 0.196 | 43.000 |
| ResNet-50 L2 eps 0.0 | 0.086 | 0.654 | 0.178 | 43.333 |
| mnasnet1_0 | 0.092 | 0.646 | 0.189 | 44.333 |
| vgg11_bn | 0.106 | 0.635 | 0.193 | 44.667 |
| InfoMin: ResNet-50 | 0.086 | 0.659 | 0.168 | 45.333 |
| vgg13_bn | 0.101 | 0.631 | 0.180 | 47.000 |
| mnasnet0_5 | 0.110 | 0.617 | 0.173 | 51.000 |
| MoCo: ResNet-50 | 0.107 | 0.617 | 0.149 | 53.000 |
| alexnet | 0.118 | 0.597 | 0.165 | 53.333 |
| squeezenet1_1 | 0.131 | 0.593 | 0.175 | 53.667 |
| PIRL: ResNet-50 | 0.119 | 0.607 | 0.141 | 54.667 |
| shufflenet_v2_x0_5 | 0.126 | 0.592 | 0.160 | 55.333 |
| InsDis: ResNet-50 | 0.131 | 0.593 | 0.138 | 56.667 |
| squeezenet1_0 | 0.145 | 0.574 | 0.153 | 57.000 |

Table 3: Benchmark table of model results for highest out-of-distribution robustness, aggregated over all 17 datasets from Geirhos et al. (2021).

| model | OOD accuracy ↑ | rank ↓ |
|---|---|---|
| ViT-22B_ft-224 (4B) | **0.837** | **1.000** |
| Noisy Student: ENetL2 (300M) | 0.829 | 2.000 |
| ViT-22B_ft-384 (4B) | 0.798 | 3.000 |
| ViT-L (14M) | 0.733 | 4.000 |
| CLIP: ViT-B (400M), 80 prompts | 0.708 | 5.000 |
| Imagen (860M) | 0.706 | 6.000 |
| ViT-L | 0.706 | 7.000 |
| SWSL: ResNeXt-101 (940M) | 0.698 | 8.000 |
| BiT-M: ResNet-152x2 (14M) | 0.694 | 9.000 |
| StableDiffusion | 0.689 | 10.000 |
| BiT-M: ResNet-152x4 (14M) | 0.688 | 11.000 |
| BiT-M: ResNet-101x3 (14M) | 0.682 | 12.000 |
| BiT-M: ResNet-50x3 (14M) | 0.679 | 13.000 |
| SimCLR: ResNet-50x4 | 0.677 | 14.000 |
| SWSL: ResNet-50 (940M) | 0.677 | 15.000 |
| BiT-M: ResNet-101x1 (14M) | 0.672 | 16.000 |
| ViT-B (14M) | 0.669 | 17.000 |
| ViT-B | 0.658 | 18.000 |
| BiT-M: ResNet-50x1 (14M) | 0.654 | 19.000 |
| SimCLR: ResNet-50x2 | 0.644 | 20.000 |
| ViT-22B_ft-560 (4B) | 0.639 | 21.000 |
| densenet201 | 0.621 | 22.000 |
| densenet169 | 0.613 | 23.000 |
| SimCLR: ResNet-50x1 | 0.596 | 24.000 |
| resnext101_32x8d | 0.594 | 25.000 |
| resnet152 | 0.584 | 26.000 |
| wide_resnet101_2 | 0.583 | 27.000 |
| resnet101 | 0.583 | 28.000 |
| ViT-S | 0.579 | 29.000 |
| densenet121 | 0.576 | 30.000 |
| MoCoV2: ResNet-50 | 0.571 | 31.000 |
| inception_v3 | 0.571 | 32.000 |
| InfoMin: ResNet-50 | 0.571 | 33.000 |
| resnext50_32x4d | 0.569 | 34.000 |
| wide_resnet50_2 | 0.566 | 35.000 |
| resnet50 | 0.559 | 36.000 |
| resnet34 | 0.553 | 37.000 |
| ResNet-50 L2 eps 0.5 | 0.551 | 38.000 |
| CLIP: ViT-B (400M), 1 prompt | 0.550 | 39.000 |
| ResNet-50 L2 eps 1.0 | 0.547 | 40.000 |
| vgg19_bn | 0.546 | 41.000 |
| ResNet-50 L2 eps 0.0 | 0.545 | 42.000 |
| ResNet-50 L2 eps 3.0 | 0.530 | 43.000 |
| vgg16_bn | 0.530 | 44.000 |
| mnasnet1_0 | 0.524 | 45.000 |
| resnet18 | 0.521 | 46.000 |
| mobilenet_v2 | 0.520 | 47.000 |
| MoCo: ResNet-50 | 0.502 | 48.000 |
| ResNet-50 L2 eps 5.0 | 0.501 | 49.000 |
| vgg13_bn | 0.499 | 50.000 |
| vgg11_bn | 0.498 | 51.000 |
| PIRL: ResNet-50 | 0.489 | 52.000 |
| mnasnet0_5 | 0.472 | 53.000 |
| InsDis: ResNet-50 | 0.468 | 54.000 |
| shufflenet_v2_x0_5 | 0.440 | 55.000 |
| alexnet | 0.434 | 56.000 |
| squeezenet1_1 | 0.425 | 57.000 |
| squeezenet1_0 | 0.401 | 58.000 |

