# OpenReview forum: "Intriguing Properties of Generative Classifiers"
_ICLR.cc/2024/Conference — ICLR 2024 spotlight_

### Official Review · Reviewer_QCiG · 2023-10-31

**Soundness:** 4 excellent
**Presentation:** 3 good
**Contribution:** 3 good
**Rating:** 8
**Confidence:** 3

**Summary:**

The rebuttal addresses my concerns well so I raised my score.

---

The tradeoff between generative and discriminative classifiers is a long-running issue in ML research (the earliest that I can recall is Andrew Ng's paper in the early 2000s).  It's generally been understood that generative classifiers can be more robust and more sample efficient but suffer if the generative model is mis-specified or under-powered.  Due to progress in the quality of generative models, it is worthwhile to revisit this tradeoffs, and it seems that generative classifiers now have realized these substantial advantages in robustness.  This paper is a useful contribution to this ongoing research trend by performing more detailed analysis on the inductive biases of generative classifiers.

notes:
  -Discriminative vs. generative classifier analysis.
  -Analysis is that generative classifiers have much higher shape bias, good OOD accuracy, alignment with human errors, and understand some perceptual illusions.
  -Generative classifier adds noise and reconstructs with the class label as the prompt, and then the reconstruction error is used to classify.

**Strengths:**

-The findings of this paper are interesting, and shows the power of generative classifiers in OOD accuracy as well as consistency with human classification.
  -The discussion is generally interesting, and I liked the idea that deficiencies of generative models could be probed through this approach (for example the inability to recognize rotated images as well as humans).
  -The study on the duck-rabbit and old woman - young woman illusions are nice.

**Weaknesses:**

-This is an analytical study of an existing approach with existing methods.  The general direction of the findings is also not terribly surprising, although the exact degree of the results and the precision of this study is still useful I think.

**Questions:**

-Do you think it would be possible to distill the generative classifier into a much stronger discriminative classifier, so that the computational benefits could be achieved while improving robustness and other properties like shape bias?

-Have you thought about probing to find places where discriminative classifiers have an advantage over generative classifiers?  Conceptually, I might try to think about types of images or classes which generative models are currently unable to generate well.

---

> ### Author Response · Authors · 2023-11-17
> **Author response**
>
> Dear reviewer QCiG,
>
> We’d like to thank you for your time and review. We’re happy that you appreciated the historical context of generative vs. discriminative classifiers and found our approach a “**useful contribution**” with “**excellent soundness**” and “**interesting findings & discussion**”. We address the questions and concerns you raised below.
>
> 1. *“The general direction of the findings is not terribly surprising, although the exact degree of the results and the precision of this study is still useful I think”:*
>
> We are happy to hear that the reviewer appreciated the “precision of the study” we undertook in this paper. We believe (as also mentioned by reviewer e2eW) that “the significance of the empirical results are the key novelty of the paper”. In particular, we’d like to point out a few aspects where our empirical results are a significant and novel contribution, which can be assessed independently of prior expectations:
> - This is the first study to compare modern generative and discriminative classifiers with large-scale human perceptual data on 17 challenging datasets
> - The method sets a new SOTA in shape bias (first-ever human-like shape bias by a ML model)
> - We demonstrate that generative classifiers achieve SOTA in human error alignment, even though deep learning models of human visual perception (and deep learning models in general) are currently dominated by discriminative classifiers
> - We expect a number of different follow-up directions will emerge from this work: in the ML community, harnessing generative pre-training in combination with faster inference (perhaps even via distillation as you suggested?) for various downstream tasks; in the cognitive science community, studying perceptual illusions with generative models; in the neuroscience community, building more accurate models of how generative and discriminative processes might be combined in biological systems like the primate brain.
>
> 2. *Distillation:*
>
> That is a very interesting suggestion! Yes, we think it might be possible to distill the scores (i.e., variational lower bound estimates) from Imagen into a discriminative classifier. Next to the computational cost of generating thousands of samples, another practical challenge might be the architectural mismatch between the diffusion model U-Net and most discriminative models; for example a diffusion model has cross attention between the input text and image while a two tower model like CLIP does not. So we are not sure how well it would work in practice, but think it would be an interesting direction to explore in future work. More broadly, our experiment with adding diffusion-style noise to a supervised classifier (Section 4) shows that lessons from diffusion models can be used to improve the shape bias of a discriminative model; which is in line with https://arxiv.org/abs/2304.08466 using a combination of synthetic and natural images to improve ImageNet accuracies.
>
> 3. *Probing such that discriminative classifiers have advantage over generative classifiers:*
>
> We note that our experimental evaluation is unbiased since we are using a very broad existing benchmark of 17 datasets that is, in fact, a benchmark designed for discriminative models and not generative models. The results presented in the manuscript are, therefore, in no way cherry-picked in favour of generative models.
>
> Within our experiments, we already see some places where discriminative models are better than generative counterparts: for example, as we observed in Section 3.2 and discuss in Section 5, generative models have a hard time classifying or generating rotated images while discriminative models are easily able to do this. Similarly, in the shape-vs-texture bias experiment, if the objective is to recognize the correct texture, discriminative models have a greater advantage over discriminative models. Finally, as mentioned in Jaini & Clark, 2023, on OCR like datasets (e.g. MNIST, and SVHN), discriminative models are able to achieve near-perfect accuracy while generative models still struggle (Stable Diffusion gets ~18% and Imagen gets ~79% on MNIST).
>
> Thanks again for your questions and thoughts!

---

### Official Review · Reviewer_tjak · 2023-11-01

**Soundness:** 2 fair
**Presentation:** 4 excellent
**Contribution:** 4 excellent
**Rating:** 8
**Confidence:** 4

**Summary:**

This submission compares recently proposed generative image classifiers to human perceptual decisions. It builds on a previous work (Clark & Jaini, 2023) which suggested employing image generation models as "analysis-by-synthesis" classifiers: Each test image is classified by attempting to reconstruct a noised version of it, conditioned on different classes. The class that best supports the reconstruction is chosen as the classifier's prediction. This approach is similar to that proposed by Schott, Rauber, Bethge & Brendel (ICLR 2019), but it leverages state-of-the-art generative AI to form generative classifiers, instead of relying on the non-scalable solution of class-specific variational autoencoders.

In the current work, three such generative classifiers (based on Stable Diffusion, Imagen, and Parti) are compared to human decisions through evaluation on out-of-distribution challenges published by Geirhos, et al., NeurIPS 2021. In addition, the ability of the classifiers to comprehend bi-stable stimuli is assessed. The experiments find high human-alignment of the proposed generative classifiers, rivaling almost all previously tested models.

**Strengths:**

1. The task of comparing contemporary generative classifiers with discriminative classifiers in terms of human alignment is both important and timely. Generative modeling might be key to emulating human perceptual behavior.
2. Overall, the evaluation is well-executed and includes a wide array of benchmarks.
3. Including three different generative classifiers enhances the generality of the results.
4. The new experiments using ambiguous stimuli are both creative and compelling.
5. The control analysis, in which a ResNet is trained with additive image noise, sheds light on confounding factors in play.

**Weaknesses:**

1. My primary concern is whether the alignment between the generative models and human behavior should be attributed to the models' generative nature or their rich semantic representations. In this context, a particularly relevant comparison is between CLIP and Stable Diffusion. Both models employ the same semantic representation, while CLIP is discriminative and Stable Diffusion is generative. However, CLIP demonstrates better OOD accuracy and error consistency than Stable Diffusion. Stable Diffusion outperforms CLIP as a human perception model only in the shape-bias benchmark. However, as suggested by the control experiment, this effect might be attributed to the high-frequency content of the additive noise.
For Imagen, there is no corresponding discriminative counterpart; such a counterpart would require training a network to map images to T5-XXL encodings. Given the absence of such a control, Imagen's alignment with human perception does not provide strong evidence in favor of generative models over discriminative models.
In summary, the reported results do not conclusively indicate whether human alignment necessitates generative modeling or merely rich semantic representation. The authors could strengthen the submission by addressing this issue, either through new experiments or by revising the conclusions and abstract to reflect the existing uncertainty.

2. The clustering analysis is misleading, since CLIP has better error consistency than most of the generative classifiers. I would omit this analysis completely.

3. The lack of an open code repository undermines the transparency and potential impact of the submission.

**Questions:**

- The loss guiding the classification score (Eq. 2) averages the error over the entire diffusion process. How does the model behavior compare if only the final, resulting image is considered?
- The authors may wish to cite Golan, Raju & Kriegeskorte (2020, PNAS), which showed evidence of greater human alignment of generative classifiers over discriminative classifiers in simpler image domains.

---

> ### Author Response · Authors · 2023-11-17
> **Author response**
>
> Dear reviewer tjak,
>
> Thank you for your review and comments / suggestions! We’re happy to hear you found the presentation and contribution “**excellent**” and the approach “**well-executed**” and “**both important and timely**”.
>
> 1. *Rich semantic representation:*
>
> We agree that attributing findings to the models' generative nature or their rich semantic representations is interesting! We first note that CLIP is a contrastive based method that was trained to match images with captions; thus we don’t consider CLIP to be a discriminative model baseline; it was not trained for specific downstream tasks like object classification, but instead performs zero-shot classification. CLIP performs worse than Stable Diffusion in OOD accuracy and similarly in error consistency when CLIP uses a single text prompt rather than an ensemble of 80 different prompts. To make this clearer, we have updated Table 1 to include 1-prompt CLIP results which are directly comparable with the 1-prompt results for all generative classifiers in the paper.
>
> By semantic representation, we assume you’re referring to the text representations? If so, this is unlikely to explain the high performance of the generative classifiers. While using a powerful text model may be important in settings involving complex language descriptions, the input prompts we use are very simple. It doesn’t seem likely to us that a 10b-parameter text encoder like T5-XXL offers much of an advantage when the prompt is “a bad photo of a dog.” In other words, the challenge of the out-of-distribution datasets we use is in the visual understanding needed to solve the tasks, not in language understanding. That said, we agree that a perfect comparison would include the same language encoder for every model, thus we have added the following to the paper’s limitations section: “Furthermore, different models use different language encoders which may be a confounding factor”.
>
> 2. *clustering analysis*:
>
> Indeed, our Figure 6 showed CLIP in the ‘model cluster’ rather than the ‘human cluster’ because we sorted the models manually and admittedly ad-hoc by model type, starting with CNNs etc. In response to your concern, we have updated the plot with a more principled sorting method: we now sort by average error consistency with human observers - thus the resulting clusters are now based on actual data, rather than manual sorting. There are still broadly two clusters that emerge - one big cluster encompassing most models, and the other encompassing humans, generative classifiers, CLIP and a few other models. Consequently, we have also updated the annotation from “Generative classifiers are in the human cluster, not the model cluster” to “Generative classifiers show high consistency with humans and lower consistency with ‘traditional’ models” (since it’s no longer just a ‘human cluster’). We believe the updated plot (Figure 6) better reflects the empirical finding, thanks for your comment!
>
> 3. *Open sourcing code:*
>
> We agree with the reviewer and are working on open-sourcing code as soon as possible. We would additionally like to note that most components of our paper can be replicated already using existing open-source repositories by various different authors and repos:
>
> The model-vs-human toolbox that contains all datasets, benchmarks, and various models: https://github.com/bethgelab/model-vs-human
>
> Zero-shot classification code for Stable Diffusion: https://github.com/diffusion-classifier/diffusion-classifier
>
> Finally, we have provided a high level of detail for reproducibility. A core experiment from our study (increased shape bias from diffusion-style training) has already been successfully replicated by an independent third party researcher based on our methods section. We provide the link to this independent replication with the caveat that they link to our publicly available non-anonymous version of the paper that might compromise the double-blind review process. Please do not visit the link if that is a concern: https://github.com/paulgavrikov/diffusion-noise-cls-pytorch
>
>
> 4. *Only considering final, resulting image:*
>
> Maybe there is a misunderstanding: in our experiments we always evaluate the score based on the final denoised image (for example please see Figure 3 in Clark & Jaini 2023 on which our approach is based). We also do not perform sampling but just run the forward diffusion process to get scores. The averaging in Eq.2 is over the noise level -  we sample multiple noise levels to noise the target image and then denoise it to get the relevant scores. We have added an algorithm box in Appendix A to make this clearer.
>
>
> 5. *Citing Golan, Raju & Kriegeskorte (2020):*
>
> We have added a citation to the controversial stimuli paper in Section 3 (“These findings appear consistent with the MNIST results by Golan et al. (2020) reporting that a generative model captures human responses better than discriminative models.”)
>
> Thanks again for your detailed review!

---

> > ### Comment · Reviewer_tjak · 2023-11-20
> > **Response to author response**
> >
> > Thank you for your replies.
> >
> > *1. Rich semantic representation:*\
> > Indeed, by "semantic" I mean textual description. Whether CLIP qualifies as a discriminative model is an interesting discussion (I would argue that CLIP is indeed discriminative); however, we likely agree that CLIP isn't a generative model. The advantage of stable diffusion over single prompt CLIP supports your conclusion; it shows a benefit for a generative model over a comparable model that maps images to text descriptions.
> >
> > However, as I mentioned in my previous review, we don't have a parallel non-generative control for Imagen. You noted that the test prompts are simple, thus minimizing the text encoder's impact. Yet, the text encoder also influences the training of the model. If the encoding of the training image descriptions is significantly superior, this might result in a better implicit visual representation. Given the impracticality of conducting an experiment to rule out this alternative interpretation within the remaining time---it would require training a model mapping images to T5-XXL embeddings of textual descriptions---I suggest addressing the remaining uncertainty about generative modeling versus merely learning to associate images and textual descriptions in the discussion.
> >
> > *4. Only considering final, resulting image:*\
> > It was indeed my misunderstanding. It is now clear.

---

> > > ### Author Response · Authors · 2023-11-22
> > >
> > > Thanks for your fast response! We will update the discussion to reflect the uncertainty regarding generative modeling vs. text encoding for Imagen.

---

### Official Review · Reviewer_e2eW · 2023-11-02

**Soundness:** 4 excellent
**Presentation:** 4 excellent
**Contribution:** 3 good
**Rating:** 8
**Confidence:** 4

**Summary:**

The paper revisit the idea of generative classifiers, which performs classification based on comparing class-conditional likelihood of a generative model, particularly focusing on the recent state-of-the-art generative models such as Imagen, Stable Diffusion, and Parti. The paper conducts a set of extensive comparison between discriminative vs. generative modeling paradigm, including comparisons in terms of shape bias, out-of-distribution accuracies, and error consistency, largely adopting the setups from Geirhos et al., 2021. The experimental results highlight that generative classifiers with state-of-the-art models are now capable, unlike previous attempts, to achieve human-like shape bias and OOD-accuracy, also advancing the error consistency with respect to humans.

**Strengths:**

Overall, the presentation is very clear, and the manuscript is well-written. The introduction section nicely summarizes the historical and related literature on generative classifiers, which can help the readers. I think the significance of empirical results are the key novelty of the paper. As an additional point, the method considered in this paper is simple and easy-to-implement.

**Weaknesses:**

I do not see major weaknesses from the paper. Although I do not much concern about it, some readers may do regarding the lack of technical novelties, considering the previous methods on generative classifiers. As a minor point, I think a more discussion with "ViT-22B-384 trained on 4B images", given that this discriminative model is still fairly possible to be considered as a competitive state-of-the-art compared to the generative modeling results that the paper highlights.

**Questions:**

- Technically, it seems the paper applies different resizing resolutions for different generative models considered - e.g., 64x64 for Imagen and 512x512 for Stable Diffusion. I would like to see a discussion on the more reasons for this and its effect to the final performance.
- If possible, it would be helpful for the readers if there could be any discussion on practical considerations and design choices on the proposed method. It seems the current version ended up with computing pixel-wise mean-squared errors to measure conditional likelihoods, but wonder if there was other considerations in attempts to improve the performance of generative classifier. Such an ablation study would help for the future research on developing generative classifiers.

---

> ### Author Response · Authors · 2023-11-17
> **Author response**
>
> Dear reviewer e2eW,
>
> Thanks for your time and thorough review, we’re happy to hear that you appreciated the “**extensive comparison**”, the paper’s “**excellent**” soundness and presentation, and the “**significance of the empirical results**”. We address the questions and concerns you raised below.
>
> *More discussion regarding ViT-22B-384:*
>
> That’s a good point, ViT-22B-384 is indeed a very competitive classifier (perhaps not surprising given that it is a large vision transformer with 22B parameters trained on 4B images for object recognition). In response to your comment, we’ve now added some discussion and made sure to reference the model in each of the empirical results sections (see 3.1, 3.2 and 3.3). For instance, in Section 3.2, we now write *“We find that Imagen and Stable Diffusion achieve an overall accuracy that is close to human-level robustness (cf.  Figure 3) despite being zero-shot models; these generative models are outperformed by some very competitive discriminative models like ViT-22B achieving super-human accuracies.”*; Section 3.1 states *“we report that all three generative classifiers significantly outperform ViT-22B (Dehghani 2023), the previous state-of-the-art method in terms of shape bias, even though all three models are smaller in size, trained on less data, and unlike ViT-22B were not designed for classification.”* and Section 3.3 states *“While a substantial gap towards human-to-human error consistency remains, Imagen shows the most human-aligned error patterns, surpassing previous state-of-the-art (SOTA) set by ViT-22B, a large vision transformer (Dehghani 2023).”*
>
>
> *Different resizing resolutions:*
>
> Imagen uses lower input resolution than Stable Diffusion because Imagen is a cascaded diffusion model and we only use the first stage image generator, not the superresolution models. The low resolution certainly is a disadvantage for Imagen in terms of performance, which we think makes its strong results all the more impressive. We have added the following note in the corresponding methods section on preprocessing (Section 2) to explain this choice: “ We preprocess the 17 datasets in the model-vs-human toolbox by resizing the images to $64 \times 64$ resolution for Imagen, $256 \times 256$ for Parti, and $512 \times 512$ for SD since these are the resolutions for the each of the base models respectively.”.
>
> *Design choices:*
>
> We used exactly the same design choices as mentioned in Clark & Jaini (2023) for our experiments. We have now added a discussion on this in Appendix A.
>
> *L2 loss function:*
>
> We use the L2 loss function for diffusion-based models since it approximates the diffusion variational lower bound (see Eq. 2) and thus results in a Bayesian classifier. Furthermore, both Stable Diffusion and Imagen are trained with the L2 loss objective. Thus, a different loss function will no longer result in a Bayesian classifier and will not work well due to differences from the training paradigm. We have added a discussion on the motivation for this choice in Appendix A.
>
> Thanks again for your questions and time!

---

### Meta-Review · Area_Chair_q8j9 · 2023-12-14

**Metareview:**

The paper explores the effectiveness of generative classifiers in object recognition. It highlights four key properties: high human-like shape bias, near-human out-of-distribution accuracy, alignment with human error patterns, and understanding perceptual illusions. During the review process, the authors effectively addressed concerns about technical novelty and methodology, expanding discussions on competitive models and clarifying the role of semantic representations. The post-rebuttal improvements significantly enriched the paper, particularly in discussing the generative nature versus rich semantic representations, and acknowledging uncertainties and limitations, thereby solidifying its contributions to understanding generative classifiers in the context of human cognition. All reviewers provided very positive feedback. AC recommends accept.

**Justification For Why Not Higher Score:**

There remain some uncertainties, particularly regarding the distinction between the effects of generative nature and rich semantic representations of the models. Ambiguity like this, while acknowledged, suggests that the work could benefit from further research to solidify its findings. Also, while rigorous in its scientific inquiry, this paper does not extensively explore the practical implications of its findings. Discussions on how these generative classifiers can be utilized in real-world applications or their potential impact on the field are somewhat limited.

**Justification For Why Not Lower Score:**

Strong empirical results, thorough comparative analysis, clarity and quality of presentation and positive reception in rebuttal, lead to this decision.

---

### Decision · Program_Chairs · 2024-01-16

Accept (spotlight)